# STeBen: Steiner Tree Problem Benchmark for Neural Combinatorial Optimization on Graphs

## Abstract

The Steiner Tree Problem (STP) is an NP-hard combinatorial optimization problem with applications in areas like network design and facility location. Despite its importance, learning-based solvers for STP have been hindered by the lack of large-scale, diverse datasets necessary to train and evaluate advanced neural models. To address this limitation, we introduce a standardized dataset comprising over a million high-quality instances with optimal solutions, spanning various problem sizes and graph structures. Our dataset enables benchmarking of neural combinatorial optimization methods across both supervised and reinforcement learning paradigms, encompassing autoregressive and non-autoregressive inference approaches. Our experiments show that supervised learning excels in in-distribution settings, while reinforcement learning generalizes better to unseen problem sizes, highlighting a trade-off between solution quality and generalization. We compare NCO methods across different STP scales and graph types, and demonstrate that solvers trained on our datasets generalize well to real-world instances without fine-tuning, proving its practical utility. We hope this benchmark promotes further STP research and advances NCO techniques for broader combinatorial optimization challenges.

## 1 Introduction

The Steiner Tree Problem (STP) is an NP-hard combinatorial optimization challenge focused on finding the minimum-cost tree spanning a set of nodes (terminals) in a weighted graph. Solving the STP is essential for optimizing connections between multiple objects or locations while minimizing costs or resources. It has diverse applications, including circuit (MacGregor Smith and LIEBMAN, 1979) and network design (Gouveia and Magnanti, 2003), facility location (Ljubić, 2007), phylogenetics (Lu et al., 2003), and image processing (Russakovsky and Ng, 2010). The STP has many variant extensions with unique characteristics such as the number of nodes, edge weight distribution, graph structure, and other specific constraints.

Due to its significance, there are many algorithms for the STP, which can be broadly categorized into two types: classical rule-based heuristics (Esbensen, 1995) and those solved using Mixed Integer Linear Programming (MILP) (Gamrath et al., 2017). Rule-based heuristics tend to be specialized for specific scenarios, and designing such heuristics is highly non-trivial. Exact MILP solvers are more versatile but generally suffer from poor scalability. These challenges of the classical algorithms are not unique to STP but are common in general combinatorial optimization (CO) domains. To address these challenges, recent research has focused on Neural Combinatorial Optimization (NCO), leveraging neural networks to enhance solution methods. However, NCO research has mainly concentrated on a few CO tasks, such as the Traveling Salesman Problem (TSP), Capacitated Vehicle Routing Problem (CVRP), and Maximum Independent Set (MIS).

Learning-based solvers for combinatorial optimization problems can generally be classified by their solution generation approach and learning paradigm. Constructive solvers generate a single solution in one pass, while improvement solvers iteratively refine solutions through local search techniques (Chen and Tian, 2019; Li et al., 2021; d O Costa et al., 2020; Wu et al., 2021; Hou et al., 2022). Given the importance of generating solutions efficiently with minimal prior knowledge, our benchmark focuses on constructive solvers. These methods have shown promise for various CO problems, efficiently

generating solutions in a single forward pass and minimizing the need for extensive human expertise. However, research for the STP has been limited due to the lack of large-scale, high-quality datasets required for training sophisticated models. Existing benchmarks, such as SteinLib (Koch et al., 2001), provide only a few dozen instances per scenario, which are insufficient for advancing state-of-the-art machine learning methods. In contrast, problems like the vehicle routing problems have benefited from vast, well-organized datasets, enabling more effective training and evaluation.

To address the limitations in STP research due to the lack of large-scale datasets, we introduce **Ste**iner Tree Problem **Ben**chmark (**SteBen**), a standardized dataset containing over a million STP instances with exact solutions across various scenarios. Our dataset covers a wide range of graph types, including variations in terminal node counts, data sizes, and edge distributions, making it compatible with existing STP libraries. Our dataset supports the evaluation of NCO methods under both supervised and reinforcement learning frameworks, covering both autoregressive and non-autoregressive methods. To facilitate this, we provide a dataset for supervised learning and a specialized reinforcement learning environment tailored for STP, where agents can interact and learn within the same problem scenarios.

Using **SteBen**, we conducted a comprehensive comparison of NCO methods and classical algorithms, focusing on four distinct NCO groups: (1) autoregressive supervised learning, (2) non-autoregressive supervised learning, (3) autoregressive reinforcement learning, and (4) non-autoregressive reinforcement learning. Since there has been limited research on learning-based solvers specifically for the STP, we aimed to evaluate broad methodological categories rather than specific state-of-the-art (SOTA) solvers, to reveal which approaches show the most promise for solving the STP.

Our contributions can be summarized as follows:

- We provide a benchmark dataset and implemented baselines for STP to evaluate a comprehensive range of NCO methods.
- We compare NCO methods, highlighting their strengths and weaknesses on the STP across different scales and graph types.
- NCO solvers trained on **SteBen** generalize well to real-world instances without additional fine-tuning, demonstrating its practical utility.

We hope **SteBen** will drive further research into NCO methods for STP and foster the development of robust, generalizable techniques for broader combinatorial optimization challenges. By offering a comprehensive benchmark, **SteBen** aims to be a key resource for the CO community.

## 2 RELATED WORK

### 2.1 CLASSIFICATION OF CONSTRUCTIVE SOLVERS

Machine-learning based constructive solvers can be divided into autoregressive, which incrementally extend partial solutions, and non-autoregressive, which generate solutions in a single step. They can also be classified by their learning approach: supervised learning, which uses labeled data, and reinforcement learning, which explores the solution space without labels.

### 2.1.1 AUTOREGRESSIVE SOLVERS

Autoregressive supervised learning methods predict the next sequence value based on the current one. Pointer Networks (Meire et al., 2015) introduced this idea, predicting sequences step-by-step. Later improvements include dividing problems into subproblems (Nowak et al., 2018) and using a variational autoencoder to solve them in a compressed latent space (Hottung et al., 2020). Recently, Drakulic et al. (2024) applied imitation learning techniques for enhanced performance on supervised datasets. Autoregressive reinforcement learning methods also extend partial solutions iteratively. This approach, which efficiently handles large combinatorial action spaces, is widely adopted in RL-based solvers. Bello et al. (2016) applied reinforcement learning to Pointer Networks, while graph embedding networks (Khalil et al., 2017) and attention modules (Kool et al., 2018) further refined this idea, leading to improved methods like POMO (Kwon et al., 2020) and Sym-NCO (Kim et al., 2022).

### 2.1.2 Non-autoregressive Solvers

Non-autoregressive methods generate solutions in a single step, avoiding the error accumulation seen in autoregressive methods. Several studies have taken a traditional supervised learning approach (Li et al., 2018; Joshi et al., 2019; Fu et al., 2021; Geisler et al., 2021), while advanced generative models like VAEs (Hottung et al., 2020), GANs (Cheng et al., 2022; Li et al., 2022), and GFlowNets (Zhang et al., 2023) have gained traction. DIFUSCO (Sun and Yang, 2023), a GNN-based diffusion model, has shown promising results, further enhanced by cost-guided search (Li et al., 2024). Non-autoregressive reinforcement learning is less common due to the complexity of large action spaces, though Qiu et al. (2022) introduced a scalable approach using parameterization and the REINFORCE algorithm.

### 2.2 Steiner Tree Problem Solvers

As mentioned in Section 1, the STP has been relatively less explored within the NCO domain. (Yan et al., 2021) utilized the Double Deep Q Network (DDQN) approach to tackle the STP in an end-to-end manner. (Zhang and Ajwani, 2022) introduced an algorithm for the STP that generates near-optimal solutions by relaxing the constraints for the Mixed Integer Linear Programming (MILP) of the STP. (Ko et al., 2023) addressed the Steiner Tree Packing Problem (STPP), which is an extended version of the STP where multiple Steiner tree problems are grouped together. Additionally, several works have focused on a variant of the STP called the Euclidean Steiner Tree Problem, in which the points are scattered on the Euclidean space, and the solution ensures that no two edges intersect (Ras et al., 2017; Wang et al., 2022; Hsu et al., 2022; Brazil et al., 2024).

### 2.3 Steiner Tree Problem Datasets

(Koch et al., 2001) generated a STP benchmark dataset. Their dataset consists of various STP instances with different levels of difficulty and their corresponding solutions. On the other hand, there was a challenge for solving STP on undirected edge-weighted graphs (Bonnet and Sikora, 2018). However, the numbers of instances in both datasets are limited to a few dozens, which is insufficient to be used to train a sophisticated neural network. There are also datasets for variant versions of STPs. Pedersen et al. (2024) published a dataset for the Quato Steiner tree problem, which additionally considers edge capacities when finding the Steiner tree. Lee et al. (2022) published Respack dataset for the STPP.

## 3 Preliminaries

### 3.1 Learning-Based Combinatorial Optimization Solvers

Let $\mathcal{F}_s$ be the set of feasible solutions for a combinatorial optimization problem (COP) instance $s \in \mathcal{S}$, where $\mathcal{S}$ is the space of all instances associated with a cost function $c_s : \mathcal{F}_s \to \mathbb{R}$. The objective is to find the optimal solution $f_s^\star$ that minimizes the cost for a given instance $s$:

$$f_s^\star = \arg\min_{f \in \mathcal{F}_s} c_s(f). \tag{1}$$

A learning-based combinatorial optimization solver is a parameterized algorithm $\hat{f}_\theta$ that aims to approximate the optimal solution for any given instance $s$. The goal is to learn parameters $\theta \in \Theta$ such that $\hat{f}_\theta(s) \approx f_s^\star$ for all $s \in \mathcal{S}$.

Learning-based solvers are categorized according to their learning paradigm, with detailed explanations of each approach provided in Appendix B. To tackle the challenges in complex combinatorial optimization problems, it is essential to have access to high-quality datasets and environments that capture the diverse and intricate nature of these instances. A standardized, large-scale dataset with exact solutions enables the training and evaluation of advanced neural models under both supervised and reinforcement learning paradigms. Additionally, providing specialized environments for reinforcement learning facilitates the development of algorithms capable of handling the intricacies of such problems.

## 3.2 THE STEINER TREE PROBLEM

An instance of the **Steiner Tree Problem (STP)** can be defined within this framework. The STP instance is given as an undirected graph $s = (V, E)$, where $V$ is the set of vertices and $E$ is the set of edges, along with a subset of vertices $T \subseteq V$ called *terminals*. Each edge $e \in E$ has an associated non-negative cost $c_s(e)$.

The goal is to find a tree $f = (V', E') \in \mathcal{F}_s$ that spans all the terminals in $T$, where $V' \subseteq V$ and $E' \subseteq E$, such that the sum of the edge costs is minimized:

$$f_s^\star = \arg\min_{f \in \mathcal{F}_s} \sum_{e \in E'} c_s(e). \tag{2}$$

We remark that the STP is NP-hard, making it infeasible to find exact solutions for large instances. Therefore, heuristic and approximation algorithms are usually employed to solve the STP, as in other combinatorial optimization problems.

As illustrated in Figure 1, we compare the STP with the Traveling Salesman Problem (TSP), a well-known combinatorial optimization problem, using a toy instance. While relocating a node in the TSP leads to minor changes in the optimal tour, adjusting the position of a terminal in the STP significantly impacts the selected edges and can even alter the total number of edges in the optimal Steiner tree. This demonstrates that dependencies in the STP may extend over relatively distant regions, and small perturbations can lead to substantial variations in the solution space.

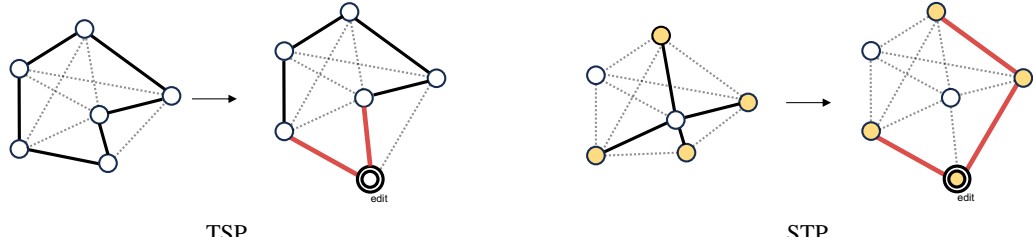

TSP                    STP

Figure 1: Comparing (a) TSP and (b) STP: A toy example illustrating the change in solution distribution due to a small perturbation (Yellow nodes: terminals; Bold edges: unchanged solution edges; Red edges: changed solution edges).

Effectively applying learning-based solvers to the STP requires addressing its unique challenges, such as the significant impact of small perturbations on the solution space and the complex dependencies across the graph.

## 4 STEBEN: A STEINER TREE PROBLEM BENCHMARK FOR NCO

### 4.1 DATASET

To address the challenges in complex combinatorial optimization problems, it is crucial to have access to high-quality datasets that capture the diverse and intricate nature of the STP. **SteBen** provides a standardized, large-scale dataset with exact solutions, enabling the training and evaluation of advanced neural models under both supervised and reinforcement learning paradigms. Our dataset includes 1.28 million optimally solved samples from various graph models *i.e.*, Erdős-Rényi (ER), Watts-Strogatz (WS), Random Regular (RR), and Grid. Node sizes for training include 10, 20, 30, 50, and 100, while the test datasets cover not only these sizes but also larger instances of 200, 500, and 1000 nodes to test the generalization capabilities of models. Additionally, **SteBen** offers a specialized environment for reinforcement learning, facilitating the development of algorithms capable of handling the intricacies of the STP across diverse problem scales and structures.

In generating the STP instances, we followed the procedures outlined in previous works (Yan et al., 2021; Ko et al., 2023). Graphs were sampled from ER, WS, RR, and grid models. For grid-based instances, nodes have 2-dimensional location features, making them analogous to Euclidean-space combinatorial optimization problems like TSP and VRPs. In each graph, terminals were randomly

selected with a 0.2 probability, and edge costs were assigned as random integers from a truncated Gaussian distribution over $[1, 2^{16}]$. We ensured that all graphs were connected by repeating the generation process if isolated nodes were present. The complete dataset generation process is detailed in Algorithm 1 and Appendix C.

---

**Algorithm 1** Dataset Generation for STP Instances

---

**Require:** Number of instances $N$, number of nodes $n$, terminal probability $p_t = 0.2$, graph type $G \in \{\text{ER}, \text{RR}, \text{WS}, \text{Grid}\}$
1: Initialize an empty dataset $\mathcal{D}$
2: **for** $i = 1$ to $N$ **do**
3:     **while** G is disconnected **do**
4:         Generate graph $G(V, E)$ based on graph type $G$ and parameters
5:     **end while**
6:     Assign each vertex $v \in V$ to be a terminal with probability $p_t$
7:     Assign edge costs $c(e)$ from a truncated Gaussian distribution $\mathcal{N}(\mu, \sigma^2)$, truncated to $[1, 2^{16}]$
8:     Add the generated graph $G$ to dataset $\mathcal{D}$
9: **end for**
10: **return** dataset $\mathcal{D}$

---

### 4.2 BENCHMARKING BASELINES

**SteBen** offers a carefully curated set of diverse construction heuristic methods to evaluate their performance on STP. The baselines are classified into four combinations based on two characteristics: SL against RL, and Autoregressive versus Non-autoregressive. All models were trained on our benchmark and evaluated under the same computational budget, using identical feature embedding techniques and decoding strategy which is proposed by (Yan et al., 2021) to ensure feasible STP solutions (*see* Appendix K.2). We ensured a fair comparison by excluding transductive learning techniques *i.e.* active search and Monte Carlo Tree Search (MCTS).

This chapter provides a brief summary of the baselines and outlines any required modifications to implement in STP when the baseline was not initially intended for STP.

**Autogressive model and supervised learning.** Autoregressive and supervised learning approaches, such as LEHD (Luo et al., 2024) and BQ-NCO (Drakulic et al., 2024), have recently concentrated on exploiting the symmetries present in COPs. These methods reconstruct the problem by excluding selected nodes and recursively considering the remaining nodes, relying on the tail-recursive property of routing problems to enhance generalization. However, STP lacks this tail-recursive property, as the selection of remaining edges and Steiner vertices depends heavily on partial solutions, making it difficult to apply these methods.

To address the challenges posed by the STP and leverage the benefits of autoregressive and supervised learning, we propose adapting PtrNet, an autoregressive, supervised learning model. To adapt PtrNet for **SteBen**, we made several key modifications: (1) representing the tree solution sequentially using level-order tree traversal, (2) prioritizing nodes based on their minimum distance to terminals rather than lexicographic order, and (3) incorporating GNN embeddings into node features to account for missing edge cost information and better capture graph topology. Further details and ablation studies on these modifications are provided in Appendix G.

**Autoregressive Model and Reinforcement Learning.** Cherrypick (Yan et al., 2021) solves the STP using an autoregressive RL model that leverages graph embedding and deep Q-learning, optimizing a reward function to minimize tree length and favor terminal selection. Similarly, the Attention Model (AM) (Kool et al., 2018) is a neural construction heuristic for routing problems that employs an autoregressive model within a reinforcement learning framework. It uses a sequence-to-sequence architecture with an attention mechanism to construct solutions by focusing on context vectors derived from previously selected nodes and their positional embeddings. To adapt the AM for the STP, the context embedding needs to be modified to reflect the specific requirements of the STP, with details provided in Appendix H. Although recent methods like POMO (Kwon et al., 2020) and Sym-NCO (Kim et al., 2022) leverage the symmetricity of COPs based on AM, applying them to the STP is not trivial because they do not account for the symmetricity in tree-structured solutions.

**Non-autogressive model and supervised learning.** DIFUSCO (Sun and Yang, 2023) is a novel graph-based diffusion framework for solving combinatorial optimization problems. They formulate NP-complete problem as {0, 1}-vector optimization problem and leverage graph-based denoising diffusion model to generate high-quality solutions.

To incorporate additional details regarding edge cost and terminal information, we modify the initialization of edge features in the Anisotropic Graph Neural Networks of DIFUSCO's Graph-based denoising network as follows:

$$\mathbf{e}_{ij}^0 = \mathbf{W}^0[f_\theta(\mathbf{x}_t), \mathbf{W}^c\mathbf{x}_{cost}, \mathbf{W}^i\mathbf{x}_{ind}] \tag{3}$$

where $f_\theta$ is the embedding function in DIFUSCO and $\mathbf{W}^0, \mathbf{W}^c, \mathbf{W}^i$ are the learnable parameters. Furthermore, decoding strategy for STP is required to get high-quality feasible solution from generated heatmap. To maintain consistency with other baselines, we incorporate the Cherrypick decoding method used by other baselines.

**Non-autogressive model and reinforcement learning.** The Differentiable Meta Solver (DIMES) algorithm (Qiu et al., 2022) is a novel approach for tackling the scalability issues in extensive COPs. In contrast to conventional DRL techniques, which suffer from by costly autoregressive decoding and repetitive refinements, DIMES use a compact continuous space to represent the underlying distribution of potential solutions. Massively parallel sampling enables stable training and fine-tuning using the REINFORCE method, resulting in a substantial reduction in gradient variance. DIMES utilizes a meta-learning framework to initialize model parameters effectively during the fine-tuning stage. This allows it to outperform current DRL-based methods on benchmark datasets for the TSP and MIS. For adaptation into the STP domain, DIMES employs a embedding technique and decoding strategy utilized across all learning-based baselines.

## 5 BENCHMARKING EVALUATION

### 5.1 EXPERIMENTAL SETUP

The baselines are trained and evaluated on the Intel Xeon Gold 6240 CPU and 8 NVidia 3090 GPUs. The accuracy of the model is measured by validating its performance on 10,000 test samples for the in-distribution results and 500 for the out-distribution results. The networks are tested once per instance during the greedy decoding process, and the best result is selected among 32 samples in each instance for the sampling metric. The training samples are split into a 1 million training set and a 280,000 validation set for supervised learning methods. [1]

In addition to the NCO approaches, we evaluate the performance of the classical solvers. Specifically, we include the MILP-based solver SCIP-Jack(Gamrath et al., 2017) and the classical heuristic 2-approximation algorithm(Kou et al., 1981) as baselines for comparison. The evaluation metrics include the average *Gap* of the predicted solutions and the computation *Time*. The inference time is calculated as the duration spent to process all the test samples for each algorithm. Additional hyperparameters and experimental details specific to each approach are presented in the Appendix.

### 5.2 RESULTS

**RQ1: Which of the four NCO methods shows the most promise for solving the STP in in-distribution evaluations?**

To address RQ1, we performed an in-distribution evaluation of four representative NCO methods: supervised autoregressive (PtrNet), supervised non-autoregressive (DIFUSCO), reinforcement learning autoregressive (AM and CherryPick), and reinforcement learning non-autoregressive (DIMES). Our goal was to determine which of these approaches shows the most promise for solving the STP, rather than focusing solely on achieving state-of-the-art performance.

As shown in Table 1, DIFUSCO (non-AR, SL) consistently outperformed the others, particularly on larger graphs, suggesting that non-autoregressive, supervised models handle complex graph

---

[1]For the STP100, the PtrNet utilized 100,000 samples; it is robust even with the smaller number of samples

structures more effectively. However, DIFUSCO's large model size resulted in longer inference times, highlighting a trade-off between accuracy and efficiency. PtrNet (AR, SL) also performed well, especially on smaller graphs, offering faster training and inference times than DIFUSCO. This indicates that autoregressive models are efficient for smaller instances. In contrast, reinforcement learning-based methods (AM, CherryPick, DIMES) generally underperformed, likely due to their reliance on exploration, which may be less suitable for the structured nature of the STP. In conclusion, supervised models, particularly DIFUSCO and PtrNet, show the most promise for solving the STP in in-distribution settings, with DIFUSCO excelling in solution quality and PtrNet offering a faster alternative for smaller graphs.

Table 1: **In-Distribution Performance on STP in Erdős-Rényi Graph.** Both training and testing are performed on datasets generated from the same graph model and problem size. Results report the gap (%) relative to SCIP-Jack's optimal solution, and the total runtime required to solve 10,000 test samples.

| Algorithm | Type | STP10 | | STP20 | | STP30 | | STP50 | | STP100 | |
|---|---|---|---|---|---|---|---|---|---|---|---|
| | | Gap (%) | Time | Gap (%) | Time | Gap (%) | Time | Gap (%) | Time | Gap (%) | Time |
| SCIP-Jack | Exact | $0.00 \pm 0.00$ | 3m | $0.00 \pm 0.00$ | 3m | $0.00 \pm 0.00$ | 3m | $0.00 \pm 0.00$ | 5m | $0.00 \pm 0.00$ | 1h |
| 2-Approx | Heuristics | $0.41 \pm 2.24$ | 6s | $1.84 \pm 4.13$ | 8s | $2.98 \pm 4.50$ | 11s | $3.84 \pm 3.93$ | 23s | $17.29 \pm 60.48$ | 82s |
| PtrNet | AR, SL, greedy | $\underline{0.75 \pm 5.04}$ | 6s | $\underline{3.75 \pm 8.23}$ | 12s | $7.61 \pm 10.12$ | 13s | $14.73 \pm 11.63$ | 27s | $25.72 \pm 13.28$ | 2m |
| AM | AR, RL, greedy | $2.54 \pm 9.06$ | 7s | $9.13 \pm 18.69$ | 11s | $13.93 \pm 21.25$ | 14s | $17.23 \pm 14.57$ | 28s | $22.22 \pm 9.56$ | 1m |
| CherryPick | AR, RL, greedy | $8.42 \pm 23.53$ | 13m | $21.19 \pm 20.34$ | 40m | $20.50 \pm 19.74$ | 52m | $29.22 \pm 17.69$ | 1.5h | $62.17 \pm 25.39$ | 3h |
| DIMES | nAR, RL, greedy | $3.99 \pm 11.41$ | 2s | $6.86 \pm 14.10$ | 4s | $\underline{5.66 \pm 8.39}$ | 8s | $\underline{11.17 \pm 10.11}$ | 21s | $\underline{11.13 \pm 7.00}$ | 2m |
| DIFUSCO | nAR, SL, greedy | $\mathbf{0.43 \pm 4.91}$ | 1.2h | $\mathbf{0.64 \pm 4.47}$ | 1.4h | $\mathbf{1.11 \pm 6.42}$ | 1.6h | $\mathbf{0.74 \pm 2.70}$ | 1.8h | $\mathbf{0.25 \pm 0.96}$ | 2.4h |
| PtrNet | AR, SL, sampling | $\mathbf{0.03 \pm 0.51}$ | 43s | $\underline{0.36 \pm 1.53}$ | 1m | $\underline{1.15 \pm 2.50}$ | 2m | $\underline{5.17 \pm 5.15}$ | 3m | $17.29 \pm 8.71$ | 13m |
| AM | AR, RL, sampling | $3.11 \pm 10.67$ | 2m | $12.12 \pm 22.32$ | 3m | $14.41 \pm 17.80$ | 9m | $18.36 \pm 15.83$ | 17m | $22.73 \pm 10.17$ | 40m |
| CherryPick | AR, RL, sampling | $1.12 \pm 0.05$ | 30m | $6.73 \pm 13.30$ | 1h | $8.18 \pm 9.78$ | 1.3h | $15.05 \pm 12.03$ | 3h | $58.13 \pm 23.75$ | 7h |
| DIMES | nAR, RL, sampling | $0.71 \pm 3.49$ | 1m | $1.09 \pm 2.86$ | 2m | $2.33 \pm 3.77$ | 4m | $8.05 \pm 6.62$ | 11m | $\underline{11.77 \pm 6.30}$ | 64m |
| DIFUSCO | nAR, SL, sampling | $\underline{0.22 \pm 3.15}$ | 7.6h | $\mathbf{0.31 \pm 3.28}$ | 8h | $\mathbf{0.58 \pm 3.45}$ | 9.6h | $\mathbf{0.31 \pm 1.65}$ | 11h | $\mathbf{0.21 \pm 0.19}$ | 14.2h |

**RQ2: How well do these NCO methods solve problems of unseen sizes, and which are the most robust?**

In exploring RQ2, we evaluated the generalization capabilities of the models by testing them on STP instances of varying sizes that differ from those seen during training. This evaluation is crucial for learning-based solvers, as their practical utility depends on their ability to solve problems at test time that are at different scales than those encountered during training.

As shown in Table 2, we observed that DIFUSCO, despite its strong in-distribution performance, experienced a significant drop in performance when tested on out-of-distribution data with different problem sizes. In contrast, the reinforcement learning autoregressive models, specifically AM and CherryPick, demonstrated more robust performance across different scales, maintaining a relatively stable performance relative to their in-distribution results. DIMES (reinforcement learning, non-autoregressive) also showed consistent performance across different scales, although it did not fully align with the trend observed for the other models.

These findings suggest that while supervised learning methods like DIFUSCO are effective within the distribution they were trained on, reinforcement learning approaches, particularly autoregressive models, may offer greater robustness and generalization to problem sizes not seen during training. This highlights the importance of considering the scalability and generalization capabilities of NCO methods when applying them to practical problems.

**RQ3: Can models trained on our synthetic STP data effectively solve real-world cases and prove more practical than classical heuristics?**

To address RQ3, we evaluated the models on real-world instances from the SteinLib benchmark (Koch et al., 2001), which are derived from practical scenarios such as network design and VLSI design. This experiment aimed to determine whether models trained on our synthetic data can effectively solve real-world problems without any fine-tuning, thereby validating their practical utility.

The results, presented in Table 3, show that the models trained on STP50 significantly outperform the baseline heuristic (2-approximation algorithm). Notably, non-autoregressive methods like DIFUSCO (supervised learning) and DIMES (reinforcement learning) achieved superior performance compared to other baselines. This indicates that models trained on synthetic data can generalize to real-world

Table 2: **Out-of-distribution Generalization Performance Measured in Relative Gap.** This table reports the relative performance degradation of trained solvers when tested on graphs with node sizes different from the training distribution, where in-distribution performance is normalized to 1. Darker shades represent greater relative performance degradation.

| Algo. | Train Nodes | Test Nodes | | | | | |
|---|---|---|---|---|---|---|---|
| | | STP10 | STP20 | STP30 | STP50 | STP100 | STP200 |
| PtrNet | 10 | 1 ( 0.75 ) | 61.85 ( 46.39 ) | 93.05 ( 69.79 ) | 128.41 ( 96.31 ) | 131.33 ( 98.5 ) | 147.45 ( 110.59 ) |
| | 30 | 0.48 ( 3.71 ) | 1.04 ( 8.12 ) | 1 ( 7.80 ) | 8.40 ( 65.52 ) | 11.95 ( 93.18 ) | 15.98 ( 124.67 ) |
| | 50 | 0.19 ( 2.80 ) | 0.67 ( 9.91 ) | 1.85 ( 27.29 ) | 1 ( 14.74 ) | 4.22 ( 63.3 ) | 8.44 ( 124.31 ) |
| AM | 10 | 1 ( 2.15 ) | 2.88 ( 66.20 ) | 4.31 ( 9.26 ) | 6.55 ( 14.08 ) | 9.90 ( 21.29 ) | 14.27 ( 60.69 ) |
| | 30 | 0.55 ( 6.45 ) | 0.94 ( 10.95 ) | 1 ( 11.71 ) | 1.28 ( 14.98 ) | 1.80 ( 21.10 ) | 2.65 ( 31.06 ) |
| | 50 | 0.42 ( 6.85 ) | 0.60 ( 9.84 ) | 0.78 ( 12.72 ) | 1 ( 16.38 ) | 1.29 ( 21.12 ) | 1.95 ( 31.963 ) |
| CherryPick | 10 | 1 ( 8.42 ) | 2.46 ( 20.75 ) | 4.88 ( 41.12 ) | 5.45 ( 45.89 ) | 6.15 ( 51.80 ) | 6.88 ( 57.91 ) |
| | 30 | 0.23 ( 4.78 ) | 0.87 ( 17.79 ) | 1.00 ( 20.50 ) | 1.27 ( 26.00 ) | 2.10 ( 42.95 ) | 2.45 ( 50.13 ) |
| | 50 | 0.20 ( 5.75 ) | 0.59 ( 17.15 ) | 0.76 ( 22.21 ) | 1.00 ( 29.22 ) | 1.31 ( 38.21 ) | 1.52 ( 44.42 ) |
| DIFUSCO | 10 | 1 ( 0.59 ) | 19.81 ( 11.69 ) | 31.34 ( 18.49 ) | 27.80 ( 16.40 ) | 23.71 ( 13.99 ) | 43.00 ( 25.37 ) |
| | 30 | 11.26 ( 8.78 ) | 3.01 ( 2.35 ) | 1 ( 0.78 ) | 1.81 ( 1.41 ) | 20.53 ( 16.01 ) | 21.00 ( 16.38 ) |
| | 50 | 10.31 ( 8.66 ) | 4.51 ( 3.79 ) | 1.74 ( 1.46 ) | 1 ( 0.84 ) | 2.90 ( 2.44 ) | 7.24 ( 6.08 ) |
| DIMES | 10 | 1 ( 3.99 ) | 2.87 ( 11.45 ) | 4.20 ( 16.75 ) | 5.36 ( 21.39 ) | 6.39 ( 25.50 ) | 8.46 ( 33.75 ) |
| | 30 | 1.01 ( 5.70 ) | 0.98 ( 5.56 ) | 1 ( 5.66 ) | 1.36 ( 7.70 ) | 2.18 ( 12.33 ) | 3.61 ( 20.43 ) |
| | 50 | 0.58 ( 6.51 ) | 0.74 ( 8.27 ) | 0.86 ( 9.59 ) | 1 ( 11.17 ) | 1.45 ( 16.21 ) | 2.13 ( 23.80 ) |

instances, and non-autoregressive approaches, in particular, demonstrate strong practical applicability.

Table 3: **Real-world Generalization Performance of Learning-based Solvers** This table shows the performance of models trained on synthetic Erdős–Rényi graphs with 50 nodes when tested on real-world instances from the SteinLib benchmark, grouped by test node sizes. Neural solvers trained on synthetic data show competitive results, highlighting their potential for real-world applications.

| Algorithm | Type | Test Nodes | | | |
|---|---|---|---|---|---|
| | | 0 - 100 | 100 - 200 | 200 - 500 | 500 - 1000 |
| 2-approx. | Heuristic | 17.79 ± 18.38 | 22.87 ± 21.19 | 43.22 ± 27.22 | 48.06 ± 23.00 |
| Pointer Network | AR, SL | 53.78 ± 27.01 | 19.10 ± 13.01 | 33.42 ± 25.01 | 35.87 ± 11.65 |
| DIFUSCO | nAR, SL | **5.59 ± 8.81** | 18.5 ± 28.4 | **5.02 ± 8.84** | 56.53 ± 100.48 |
| Cherrypick | AR, RL | 13.72 ± 30.42 | 49.98 ± 98.75 | 8.70 ± 11.20 | 43.34 ± 56.24 |
| AM | AR, RL | 17.51 ± 15.21 | 19.67 ± 15.78 | 30.09 ± 13.47 | 27.55 ± 13.39 |
| DIMES | nAR, RL | 12.56 ± 14.39 | **17.45 ± 15.53** | 19.31 ± 27.33 | **15.92 ± 19.51** |

# 6 DISCUSSION

**Discussion about nAR settings and AR settings.** As shown in Table 1, non-autoregressive models outperform autoregressive models in solving the STP. This can be attributed to the fact that STP heavily relies on the information of the currently selected partial solution, whereas AR models may suffer from a smoothing problem when aggregating partial solution information during the sequential node selection process. Especially, the smoothing of the embedding becomes more prominent when the number of terminals is large or the solution sequence is long, leading to a decrease in representation power. This explains the more significant performance degradation of AR models on larger-scale instances. Therefore, one potential direction for improving the performance of AR models is to enhance their representation power for partial solutions.

**Training sample efficiency in supervised setting.** In NCO, SL benefits from its ability to learn quickly and reliably from high-quality labeled solutions. However, obtaining a sufficient training dataset with high-quality solutions is typically very expensive in large-scale combinatorial optimization domains due to their NP-hardness. If the dataset quantity is insufficient which means the model is trained on a limited number of instances, leading to reduced generalization for unseen instances.

| Algorithm | Number of Training samples | | | | |
|---|---|---|---|---|---|
| | 1M | 50K | 10K | 1K | 100 |
| PtrNet | 14.73 | 17.74 | 18.81 | 23.42 | 25.47 |
| DIFUSCO | 0.84 | 1.28 | 6.94 | 10.85 | 12.28 |

Figure 2: Degradation of SL with respect to number of training sample in STP50

To show the impact of training samples size on the performance gap of SL methods, we train the SL method with varying numbers of training samples. Figure 2 (a) shows the relative performance degradation of each method when using reduced training samples, compared to their respective performance with the full training set (1 million samples). As expected, the performance of both methods deteriorates as the training sample size reduces. Concretely, DIFUSCO demonstrates higher robustness and effectiveness in low-sample regimes, maintaining a smaller performance gap than the Pointer Network. However, it is noteworthy that the rate of performance decline in DIFUSCO is steeper than Pointer Network when compared to their respective peak performance with 1 million samples. This figure highlights the importance of considering sample efficiency and the trade-offs between different supervised learning methods when dealing with limited training data.

**Limitations.** State-of-the-art NCO methods require large datasets, making computational efficiency crucial for reasonable training times. Unlike Euclidean-space routing problems like TSP, efficiently supplying training data for STP on graphs poses significant engineering challenges in terms of providing data in the training loop, making it difficult to train on very large-scale problems. In addition, our STP instances do not cover the full distribution of all possible STP problems, potentially affecting performance on different distributions. Additionally, while real-world STP problems often have practical constraints, our study focuses on the unconstrained version, leaving constrained versions for future work.

## 7 CONCLUSION

In this paper, we introduced **SteBen**, a comprehensive benchmark for evaluating neural combinatorial optimization (NCO) methods on the Steiner Tree Problem (STP). Our benchmark provides extensive datasets and environments for training state-of-the-art NCO methods with exact solutions. Additionally, we have implemented various NCO algorithms and a classical heuristic, providing code that allows practical experimentation across diverse scenarios. We reported the results of applying a wide range of NCO construction heuristic methods for the STP, offering detailed comparisons and analyses of their characteristics. Our comprehensive evaluation not only highlighted the strengths and limitations of existing methods but also provided valuable insights for future research. We believe that **SteBen** will serve as a stepping stone for researchers, facilitating the development of more effective NCO methods and driving further advancements in solving complex combinatorial optimization problems.

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

## A   DATASETS & SOURCE FOR BENCHMARK EXPERIMENTS

We provide the following link to download the training and test data [2]. The datasets are stored as pickle files based on networkx[3] graph format, and for larger node sizes, the data is divided into multiple pickle files. The source code for all the baseline methods we experimented with can be found in the following GitHub repository [4]. These are distributed under an MIT license.

## B   BACKGROUND ON LEARNING-BASED SOLVERS

### B.1   SUPERVISED LEARNING AND REINFORCEMENT LEARNING IN NCO

In **supervised learning**, a model is trained on a dataset $\{(s, f_s^\star)\}_{s \in D}$ where each instance $s$ is paired with a label $f_s^\star$ representing the known optimal or high-quality solution. The model aims to imitate the optimal solution by minimizing the difference $\|\hat{f}_\theta(s) - f_s^\star\|$ under an appropriate metric $\|\cdot\|$. The training objective is to minimize the loss function:

$$\mathcal{L}(\theta) = \sum_{s \in D} \|f_s^\star - \hat{f}_\theta(s)\|. \tag{4}$$

In **reinforcement learning**, the solver learns autonomously using feedback from the cost function $c_s(\cdot)$, without relying on labeled data. The solver $\hat{f}_\theta(\cdot)$ is considered a policy where the instance $s$ is regarded as the state and the output $\hat{f}_\theta(s)$ as the action. After selecting an action, the solver receives a reward $-c_s(\hat{f}_\theta(s))$. The training objective is to minimize the expected cost over the distribution $\mathbb{P}(\mathcal{S})$ of instances:

$$R(\theta) = \mathbb{E}_{s \sim \mathbb{P}(\mathcal{S})}[c_s(\hat{f}_\theta(s))]. \tag{5}$$

### B.2   AUTOREGRESSIVE AND NON-AUTOREGRESSIVE SOLVERS

**Autoregressive solvers** generate a sequence of partial solutions incrementally, conditioned on previous solutions, until a feasible solution is reached. This step-by-step approach simplifies the learning process by breaking it down into manageable stages. In the context of reinforcement learning, the advantage of reducing the action space has made autoregressive methods the predominant choice.

**Non-autoregressive solvers** generate the complete solution $\hat{f}_\theta(s)$ for a given problem instance $s$ in a single step. With recent progress in deep learning, particularly in models such as diffusion models that excel in high-dimensional spaces, these solvers have become increasingly popular for tackling large-scale combinatorial optimization problems. They avoid the error accumulation issue inherent in sequential generation.

## C   STP DATASET GENERATION

### C.1   INSTANCE GENERATION

To cover a diverse range of problem instances for the Steiner Tree Problem (STP), we generated graphs using four common models: Erdős-Rényi (ER), Random-Regular (RR), Watts-Strogatz (WS), and grid graphs. Each model introduces unique graph structures, allowing us to evaluate the performance of neural combinatorial optimization methods across various topologies. Additionally, we applied specific generation techniques to ensure that all graphs are connected, feasible, and suitable for training NCO models. Below, we describe the generation process for each graph type.

---

[2]https://drive.google.com/drive/folders/1j_vuK-Mhv0mGoAXgF8FNVn1onONX-34T?
usp=drive_link
[3]https://networkx.org
[4]https://anonymous.4open.science/r/steben-1471

**ER** To generate ER graphs, we sampled the probability for edge creation from a uniform distribution and used this value as the parameter for the ER model.

**RR** To generate RR graphs, we ensured that each node had the same number of neighbors by sampling the degree of the graph from a Uniform($\{3, 4, 5\}$) and using this degree value as the parameter for the RR model.

**WS** To generate WS graphs, we sampled the mean node degree $k \sim$ Uniform($\{3, 4, 5, 6\}$) and the rewiring probability $\beta \sim$ Uniform($0, 1$) for generating random graphs.

**Grid** When generating grid graphs, given the number of nodes $n$, we determine the grid's dimensions by sampling combinations of width and height such that their product equals $n$. For example, if $n$ is 20, we randomly sample between combinations like 4x5 or 5x4. Additionally, we ensure that at least one dimension is greater than 4. In contrast to the other graph types, for grid-based instances, we fixed all edge costs to 1. This design choice was made to ensure that the instances exhibit properties more akin to those found in Euclidean space.

## C.2 SOLVER FOR OPTIMAL SOLUTION

To facilitate supervised training, we provide both training and test sets with optimal solutions and costs to measure performance gaps against these optimal benchmarks. For this purpose, we employed the MILP-based SCIP-Jack[5] solver to compute optimal solutions for the STP instances we cover. This solver is used under the ZIB license, which allows usage by members of non-commercial and academic institutions.

## C.3 STATISTICS

The following tables describe the statistics of the training and test datasets from ER graphs. We generated 10,000 test samples for instances with fewer than 200 nodes and 1,000 samples for instances with 200 nodes or more, due to the exponentially increasing time required to find the optimal cost using MILP-based solvers as the problem scale grows. While the training data includes solutions for supervised learning, the test data only contains the optimal costs, not the solutions.

Table 4: Data statistics of training data

| Type | STP10 | STP20 | STP30 | STP50 | STP100 |
|---|---|---|---|---|---|
| # of terminals | $2.48 \pm 0.81$ | $4.13 \pm 1.70$ | $6.02 \pm 2.15$ | $9.99 \pm 2.81$ | $19.97 \pm 4.00$ |
| # of optimal edges | $2.04 \pm 1.10$ | $4.12 \pm 2.05$ | $6.60 \pm 2.62$ | $11.94 \pm 3.49$ | $26.27 \pm 5.23$ |
| # of total edges | $30.06 \pm 8.89$ | $115.98 \pm 43.35$ | $253.28 \pm 104.82$ | $686.01 \pm 312.42$ | $2663.71 \pm 1323.96$ |
| Optimal value | $53240.31 \pm 28970.65$ | $88300.01 \pm 44448.35$ | $125144.12 \pm 53819.80$ | $188669.69 \pm 67699.81$ | $310943.73 \pm 109333.33$ |
| Avg. edge cost | $32779.54 \pm 2088.35$ | $32750.46 \pm 1111.49$ | $32765.40 \pm 762.07$ | $32766.97 \pm 484.33$ | $32769.43 \pm 264.28$ |

Table 5: Data statistics of test data (from STP10 to STP50)

| Type | STP10 | STP20 | STP30 | STP50 |
|---|---|---|---|---|
| # of terminals | $2.50 \pm 0.84$ | $4.07 \pm 1.67$ | $6.00 \pm 2.17$ | $10.07 \pm 2.85$ |
| # of optimal edges | N/A | N/A | N/A | N/A |
| # of total edges | $27.96 \pm 9.98$ | $109.24 \pm 46.57$ | $241.80 \pm 111.13$ | $660.25 \pm 328.60$ |
| Optimal value | $56502.76 \pm 32505.02$ | $92096.61 \pm 48963.88$ | $130576.33 \pm 60079.15$ | $196755.18 \pm 76570.06$ |
| Avg. edge cost | $32756.82 \pm 2236.76$ | $32793.21 \pm 1171.90$ | $32774.25 \pm 823.30$ | $32760.07 \pm 520.62$ |

## D PERFORMANCE VS RUNTIME ANALYSIS

Figure 3 illustrates the tradeoff between runtime and performance (Gap %) for various algorithms, including SCIP-Jack and learning-based approaches, across different STP sizes. SCIP-Jack achieves near-optimal solutions for smaller graphs within reasonable runtime but struggles to scale efficiently

---

[5]https://scipjack.zib.de

Table 6: Data statistics of test data (from STP100 to STP1000)

| Type | STP100 | STP200 | STP500 | STP1000 |
|---|---|---|---|---|
| # of terminals | $20.00 \pm 3.95$ | $39.94 \pm 5.70$ | $99.70 \pm 8.98$ | $198.80 \pm 12.51$ |
| # of optimal edges | N/A | N/A | N/A | N/A |
| # of total edges | $2560.04 \pm 1366.75$ | $9954.42 \pm 5610.50$ | $63167.71 \pm 34243.41$ | $264444.55 \pm 136561.18$ |
| Optimal value | $322868.13 \pm 122637.03$ | $503077.40 \pm 227663.14$ | $736329.24 \pm 464148.09$ | $796172.63 \pm 785896.22$ |
| Avg. edge cost | $32767.29 \pm 278.63$ | $32762.68 \pm 147.72$ | $32769.35 \pm 61.90$ | $32771.98 \pm 39.88$ |

as graph sizes increase, with its runtime growing exponentially. In contrast, learning-based approaches such as DIFUSCO and PtrNet demonstrate better scalability to larger graphs. DIFUSCO achieves the smallest gaps consistently, albeit with higher computational cost, while PtrNet offers a balance of runtime efficiency and solution quality for smaller STPs. Reinforcement learning methods, including AM and DIMES, show moderate performance with varying tradeoffs across sizes. This analysis highlights the complementary strengths of classical and neural solvers, suggesting that the choice of algorithm depends on the runtime constraints and the scale of problems.

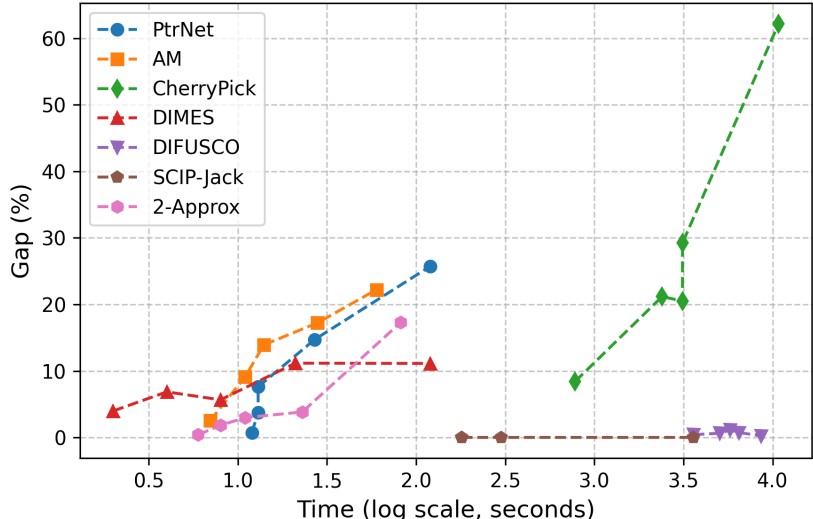

Figure 3: **Gap vs. Runtime Across STP Sizes for Different Solvers** DIFUSCO consistently achieves the smallest gaps, while PtrNet balances efficiency and quality. SCIP-Jack performs well for small graphs but struggles with scalability, and reinforcement learning methods like DIMES maintain steady performance across scales.

# E  EVALUATION ON OTHER GRAPH MODELS

Table 7 presents the evaluation results of various algorithms on different types of graphs, specifically Random-Regular (RR), Watts-Strogatz (WS), and grid graphs. The table shows the performance of each algorithm in terms of the gap (%) between the algorithm's cost and the optimal cost for different test node sizes. For grid graphs, experiments with node size 10 were excluded due to the lack of diversity in grid shapes. These experiments were conducted by training the models on instances with 50 nodes and then evaluating their performance on instances of varying scales. In the same manner as the out-of-distribution generalization evalutions in the main paper, each test instance set contained 500 samples. The results highlight interesting patterns, with NCO baselines generally exhibiting smaller gaps compared to ER graph problems. Notably, DIFUSCO consistently outperforms the heuristic solver, demonstrating superior overall performance. However, consistent with the characteristics of non-autoregressive models, both DIFUSCO and DIMES show significant performance degradation on STP10 compared to STP50, indicating weaker generalization capabilities. CherryPick displays substantial performance variation depending on the graph type, while Pointer

Network's performance sharply declines as the problem scale increases. In contrast, AM maintains a more consistent performance degradation across increasing problem scales, suggesting a more stable solution quality as the problem size grows. These results provide valuable insights into the generalization capabilities and robustness of the algorithms when applied to diverse graph structures.

Table 7: Performance Evaluation of Baseline Methods on Diverse Graph Types and Node Scales

| Algorithm | Graph Type | Test Nodes | | | | | |
|---|---|---|---|---|---|---|---|
| | | STP10 | STP20 | STP30 | STP50 | STP100 | STP200 |
| 2-Approx | RR | $0.43 \pm 1.78$ | $4.06 \pm 2.14$ | $3.14 \pm 4.44$ | $3.98 \pm 3.78$ | $5.25 \pm 3.07$ | $5.42 \pm 2.14$ |
| | WS | $0.36 \pm 2.14$ | $1.47 \pm 3.49$ | $2.77 \pm 4.33$ | $3.60 \pm 4.45$ | $3.82 \pm 3.70$ | $4.19 \pm 3.59$ |
| | Grid | - | $2.05 \pm 4.91$ | $3.24 \pm 5.27$ | $3.39 \pm 3.83$ | $3.58 \pm 2.55$ | $4.04 \pm 1.90$ |
| PtrNet | RR | $3.51 \pm 10.58$ | $3.84 \pm 7.80$ | $4.06 \pm 5.68$ | $5.74 \pm 5.17$ | $96.58 \pm 25.03$ | $83.88 \pm 22.39$ |
| | WS | $2.55 \pm 9.09$ | $4.37 \pm 8.92$ | $5.55 \pm 8.69$ | $5.74 \pm 9.88$ | $26.23 \pm 24.36$ | $49.90 \pm 37.08$ |
| | Grid | - | $7.87 \pm 13.00$ | $6.37 \pm 8.62$ | $8.07 \pm 7.16$ | $76.28 \pm 19.65$ | $79.84 \pm 16.20$ |
| AM | RR | $6.06 \pm 15.00$ | $6.44 \pm 10.80$ | $7.75 \pm 11.03$ | $8.05 \pm 6.41$ | $8.33 \pm 4.93$ | $8.55 \pm 3.63$ |
| | WS | $1.44 \pm 6.58$ | $2.34 \pm 5.67$ | $3.36 \pm 5.81$ | $3.98 \pm 4.61$ | $3.96 \pm 3.74$ | $4.27 \pm 3.59$ |
| | Grid | - | $2.12 \pm 5.33$ | $2.65 \pm 4.79$ | $3.74 \pm 4.39$ | $5.20 \pm 3.92$ | $6.95 \pm 3.27$ |
| CherryPick | RR | $10.80 \pm 24.35$ | $13.80 \pm 21.20$ | $15.48 \pm 16.75$ | $16.08 \pm 12.47$ | $17.65 \pm 10.28$ | $16.63 \pm 7.64$ |
| | WS | $27.09 \pm 48.50$ | $39.79 \pm 56.46$ | $40.01 \pm 42.65$ | $41.94 \pm 32.79$ | $39.56 \pm 31.00$ | $39.27 \pm 30.45$ |
| | Grid | - | $6.02 \pm 11.79$ | $6.74 \pm 8.64$ | $8.51 \pm 6.81$ | $7.78 \pm 4.71$ | $8.34 \pm 3.27$ |
| DIFUSCO | RR | $4.38 \pm 14.38$ | $1.80 \pm 5.74$ | $0.42 \pm 1.49$ | $0.17 \pm 0.74$ | $0.71 \pm 3.09$ | $2.23 \pm 4.34$ |
| | WS | $4.43 \pm 15.55$ | $2.53 \pm 10.28$ | $2.21 \pm 9.04$ | $1.63 \pm 7.13$ | $1.41 \pm 3.65$ | $3.15 \pm 4.85$ |
| | Grid | - | $0.06 \pm 1.48$ | $0.00 \pm 0.00$ | $0.00 \pm 0.00$ | $2.10 \pm 3.93$ | $2.93 \pm 2.98$ |
| DIMES | RR | $5.60 \pm 16.51$ | $3.38 \pm 7.884$ | $3.50 \pm 13.39$ | $2.96 \pm 3.34$ | $3.96 \pm 2.78$ | $4.60 \pm 2.11$ |
| | WS | $10.21 \pm 30.00$ | $4.18 \pm 17.44$ | $2.91 \pm 9.45$ | $2.56 \pm 3.75$ | $2.85 \pm 3.18$ | $3.86 \pm 3.36$ |
| | Grid | - | $4.05 \pm 15.39$ | $2.83 \pm 8.44$ | $3.78 \pm 2.79$ | $3.02 \pm 3.74$ | $4.96 \pm 2.19$ |

# F  CHERRYPICK

## F.1  EXPERIMENTAL DETAILS

We reimplemented the CherryPick algorithm following the original paper's description. We contacted the authors and received a portion of the code, but fully reproducing the experiments was impossible due to missing parts for the STP environment. Nevertheless, this enabled us to incorporate minor details not explicitly mentioned in the paper into our implementation. However, reproducing the bonus term for terminal selection in the reward function exactly as described in the paper and the received code was not possible. During each state transition, CherryPick updates its features, which involves calculating the shortest paths from every node to all terminal nodes that need to be connected. This process makes it challenging to efficiently train on large-scale problems due to the computational overhead.

For training CherryPick, we increased the number of STP instances used compared to the original paper, utilizing up to 100,000 instances. Due to the computational demands, only 6,000 instances were used for training on STP100. Among the hyperparameters, we used a discount factor of 0.99 instead of the 0.2 specified in the original paper, as it shows better performance.

## F.2  COMPARISON TO ORIGINAL PAPER

To validate the implementation of CherryPick, we compared our results on STP50 (fixed weight) instances with ER graphs against those reported in the original paper (show Table 8). In this setting, evaluated using the same methodology as the original paper, the gap was calculated based on the 2-approximation algorithm. Our implemented version achieved slightly lower performance compared to the original. This discrepancy can be attributed to the aforementioned challenges in perfectly replicating the reward function, as well as potential differences in the parameters that define the graph characteristics during the generation process.

Table 8: Comparison Gap with Original Paper on ER Graph

| Problem | Implemented (*ours*) | Original (Yan *et. al.*, 2021) |
|---------|---------------------|-------------------------------|
| STP50 (Fixed Weight) | $-0.29 \pm 3.38$ | $-2.5$ |

### F.3 Reward functions of STP Environment

Table 9 demonstrates the impact of including a terminal selection bonus in the reward function for CherryPick. The results indicate that incorporating a bias towards selecting terminal nodes, in addition to minimizing tree length, significantly improves performance. Specifically, the reward function with the terminal bonus consistently outperforms the one without it across all test node sizes.

Table 9: Comparison Reward Functions for Cherrypick

| Problem | Test Nodes | |
|---------|-----------|---|
| | w/ terminal bonus (original) | w/o terminal bonus |
| STP10 | $8.42 \pm 23.53$ | $20.70 \pm 38.41$ |
| STP20 | $21.19 \pm 20.34$ | $29.65 \pm 36.45$ |
| STP30 | $20.50 \pm 19.74$ | $64.78 \pm 49.23$ |
| STP50 | $29.22 \pm 17.69$ | $64.00 \pm 35.71$ |

## G Pointer Networks

### G.1 Experimental Details

We reused to the most of experimental settings as described in the original publications, including a single-layer LSTM with 512 embedding, weight initialization ranging from -0.08 to 0.08, and gradient clipping. The model was trained 300 epochs using Adam optimizer, and the best model was selected based on 1K validation set. We used 1M training instances with a learning rate of 1e-4 and and the maximum batch size allowed by the graph size. Additionally, we utilized teacher forcing, a classic technique for stable training during the early stage.

### G.2 Label Generation

To serialize nodes of the STP solution tree, we compared tree traversal methods. A general tree, where the number of children is not limited to two can be reordered using level-order, pre-order, and post-order traversal. **Level-order traversal**, also known as breadth-first search (BFS), prioritizes visiting all the children of a node before proceeding to the nodes at the next level. **Pre-order Traversal**, or depth-first search, is a method where the traversal goes deeper recursively and then visits the siblings' subtrees. Pre-order traversal prioritizes visiting the root before its children. In contrast to other two methods, **Post-order Traversal** visits all the children nodes before their respective parent node. We constructed the tree based on node indices, designating the root as the last-ordered terminal point and sorting the children in descending order. This allows us to fix the position of terminals as the latest embedding and assign consistent labels to aid in learning.

### G.3 Edge Embeddings

To compensate for missing edge cost information, we added a GNN embedding to node embedding. The embedding of node $i$ is embedded as follows:

$$\mathbf{h}_i = \mathbf{h}_i + \frac{1}{|\mathcal{N}_i|} \sum_{j \in \mathcal{N}_i} \hat{\mathbf{c}}_j \mathbf{h}_j \qquad (6)$$

where $\mathbf{h}_i$ is a node embedding of node $i$, $\mathcal{N}_i$ is a neighbor of node $i$, and $\hat{\mathbf{c}}_j$ is a normalized edge cost $(1 - \frac{\mathbf{c}_j}{\max_{i \in E} \mathbf{c}_i})$.

### G.4 ABLATION STUDIES

Table 10: PtrNet ablation experiment.

| Type | Gap (%) |
|---|---|
| PtrNet w/ Level (default.) | $0.75 \pm 5.04$ |
| PtrNet w/ Pre | $0.80 \pm 5.54$ |
| PtrNet w/ Post | $4.72 \pm 12.74$ |
| PtrNet w/ Random | $1.35 \pm 5.98$ |
| PtrNet w/ Inv. sort | $0.83 \pm 6.06$ |
| PtrNet w/o Edge Emb. | $1.13 \pm 8.28$ |

In Table 10, we represent the Gap in Erdős-Rényi graph ablation study to show the importance of each elements.

First, we explored various tree traversal methods, including Level, Pre, Post, and Random. The performance of node traversal strategies exhibits significant variance depending on the traversal method employed. The level traversal method demonstrated superior performance, while the post traversal method exhibited a notable decrease in performance compared to random ordering. In our experimental setup, where slight changes in node coordinates could lead to minor variations in indices, the post traversal method proved particularly sensitive due to its exhaustive exploration of leaf nodes, rendering the capture of such variations challenging.

Secondly, we examined the influence of node encoding order. Interestingly, encoding nodes in proximity to terminals towards the end of the sequence yielded improved performance. This observation suggests that providing crucial information about terminal-adjacent nodes just prior to the decoding phase contributes slightly to enhanced model performance.

Lastly, regarding edge embedding, despite the small node size of 10 used in this experiment, we observed a performance drop without it. These results imply that edge embedding contributes to finding the optimal solution for the STP. Therefore, advanced node embedding approaches could potentially lead to substantial performance improvements.

## H AM

### H.1 EXPERIMENTAL DETAILS

We implemented AM for the STP based on the RL4CO[6] library, because the original author's repository notes limited maintenance and suggests more recent implementations. We trained our model on 1.28 million instances per epoch, as in the original study. Unlike the routing problems in the original paper, generating the STP dataset was time-consuming, resulting in a drop in training efficiency. To calculate the advantage, we initially used the rollout baseline from the original paper, but it led to policy collapse. Therefore, we employed the mean baseline, which provided more robust training. Additionally, similar to the CherryPick paper, we used the distances to the top-K nearest terminals as node features. To reflect the problem's edge costs, we modified the AM by incorporating a process where the node embedding vectors are calculated by taking the weighted sum of the embeddings of neighboring nodes, using the edge costs as weights.

### H.2 CONTEXT EMBEDDING FOR STP

In the AM for STP, the problem state is represented using context embedding vectors from previously selected nodes and their positional encodings, guiding the next node selection based on attention

---

[6]https://github.com/ai4co/rl4co

scores. Specifically, at time $t$, the input for TSP consists of the embedding of the graph, the previous (last) node $\pi_{t-1}$, and the first node $\pi_1$. For STP, we modify this as follows:

$$h_{(c)}^{(N)} = \begin{cases} \left[ h^{(N)}, h_{\pi_{t-1}}, h_{\pi_1} \right] & t > 1 \\ \left[ h^{(N)}, \mathbf{v}^l, \mathbf{v}^f \right] & t = 1 \end{cases} \tag{7}$$

where $\mathcal{P}_t$ is the sequence of previously selected nodes up to time $t$. The embedding $h_{\pi_{t-1}}$ is the average of the embedding vectors of these nodes, providing a summary of the partial solution:

$$h_{\pi_{t-1}} = \frac{1}{|\mathcal{P}_t|} \sum_{i \in \mathcal{P}_t} h_i^{(N)}$$

Similarly, $\mathcal{T}_t$ represents the set of remaining terminals. The embedding $h_{\pi_1}$ is the average of the embedding vectors of these terminals, summarizing the remaining objectives:

$$h_{\pi_1} = \frac{1}{|\mathcal{T}_t|} \sum_{j \in \mathcal{T}_t} h_j^{(N)}$$

By adjusting the context embedding in this way, the AM can effectively handle the STP, ensuring that the embeddings reflect the current partial solution and the remaining terminals to be connected. This modification allows the model to focus on the key aspects of the STP during the construction process, leveraging similar principles to those used in solving the TSP.

# I DIFUSCO

## I.1 Experimental Details

In DIFUSCO, we primarily followed the experimental settings described in the official code, including a 12 GNN layers with 256 hidden dimension, 1000/50 for diffusion steps, 0.0002 for learning rate and 0.0001 for weight decay. We utilize the node embeddings as the distance to the terminal from each nodes, which is the method from CherryPick. The main differences are that we changed the default decoding scheme from the original Greedy decoding + 2-opt scheme to the decoding scheme used in CherryPick. Since discrete diffusion consistently outperformed continuous diffusion on the TSP and MIS datasets (Sun and Yang, 2023), we used the discrete diffusion approach in Table 1. Additionally, we employed the 10 diffusion steps for sampling.

## I.2 Edge Embedding for STP

As mentioned in 4.2, we propose a modification to the edge embedding initialization process in the Graph-based denoising network to address the limitations of the DIFUSCO approach in solving STP, where edge costs and terminal node information are crucial.

$$\mathbf{e}_{ij}^0 = \mathbf{W}^0[f_\theta(\mathbf{x}_t), \mathbf{W}^c \mathbf{x}_{cost}, \mathbf{W}^i \mathbf{x}_{ind}] \tag{8}$$

where $\mathbf{x}_{cost}$ is edge cost matrix and $\mathbf{x}_{ind}$ is indicator matrix consisting of 1's for existing edges and 0's for non-existing edges. By directly incorporating edge cost information and terminal node indicators into the edge embedding vectors, we enable DIFUSCO to effectively capture the unique characteristics of the STP and generate more accurate and cost-effective solutions.

## I.3 Ablation Studies

In this section, we conduct experiments with varying the settings of DIFUSCO. We investigate the impact of edge embeddings, diffusion model variants, and the number of GNN layers on the performance of DIFUSCO. The basic experiment setup is same as Table 2.

First, we examine the importance of edge embeddings by training a model that excludes them and only uses the node features. As expected, the results demonstrate that incorporating the actual edge

Table 11: Ablation study for DIFUSCO

| Setting | Test Nodes | | | | | |
|---|---|---|---|---|---|---|
| | STP10 | STP20 | STP30 | STP50 | STP100 | STP200 |
| DIFUSCO w/o edge embedding | $39.1 \pm 62.74$ | $63.11 \pm 52.16$ | $86.29 \pm 57.29$ | $102.34 \pm 49.82$ | $133.78 \pm 50.02$ | $190.22 \pm 76.75$ |
| DIFUSCO w/ Continuous Diffusion | $5.68 \pm 15.87$ | $3.01 \pm 8.65$ | $1.06 \pm 4.97$ | $2.50 \pm 15.72$ | $11.86 \pm 25.60$ | $28.97 \pm 30.83$ |
| DIFUSCO w/ GNN 3-layers | $2.76 \pm 7.12$ | $2.74 \pm 6.15$ | $2.39 \pm 5.14$ | $2.28 \pm 3.57$ | $2.89 \pm 2.33$ | $8.71 \pm 7.48$ |
| DIFUSCO w/ GNN 6-layers | $2.59 \pm 7.72$ | $2.13 \pm 6.00$ | $1.35 \pm 4.78$ | $0.92 \pm 3.09$ | $1.56 \pm 5.61$ | $6.99 \pm 7.27$ |
| DIFUSCO w/ GNN 9-layers | $4.33 \pm 10.95$ | $1.83 \pm 5.24$ | $0.68 \pm 2.67$ | $0.43 \pm 3.49$ | $1.01 \pm 3.54$ | $7.97 \pm 12.02$ |
| DIFUSCO | $8.66 \pm 18.5$ | $3.79 \pm 11.31$ | $1.46 \pm 5.24$ | $0.84 \pm 2.75$ | $2.44 \pm 4.48$ | $6.08 \pm 5.69$ |

costs and indicator matrix is crucial for the model's preformance. This also suggests that the features from CherryPick alone may not provide significant benefits to DIFUSCO.

Next, we compare the performance of discrete and continuous diffusion models. The experimental results align with the finding mentioned in the (Sun and Yang, 2023), indicating that discrete diffusion models generally outperform their continuous counterparts in STP, similar to TSP and MIS. The performance gap becomes more pronounced as the size of the STP instances increases.

Lastly, we investigate the impact of varying the number of GNN layers. While our model achieves good results using 12 GNN layers, consistent with (Sun and Yang, 2023), we observe an interesting phenomenon. In certain cases, models with fewer layers exhibit better performance. Specifically, reducing the number of layers leads to a decrease in overall performance but an increase in generalization power. However, as the problem size grows, models with more layers tend to perform better. This highlights the importance of selecting an appropriate number of layers based on the size or complexity of the problem.

## J  DIMES

### J.1  EXPERIMENTAL DETAILS

In DIMES, we reduced the size of the network to 6 GNN layers with 16 hidden dimensions, with an outer learning rate of 0.0005 and an inner learning rate of 0.005 to ensure the stability of the training process. The remaining hyperparameters are unchanged, including the outside steps of 120, an inner sample size of 100, and the inner steps of 15. These values remain constant regardless of the number of nodes.

The main differences are in the node embedding and decoding algorithm. The initial node embedding in the TSP consists of $(x, y)$ vectors representing all of the points. Nevertheless, we utilize the node embedding from the CherryPick approach, which utilizes the distance to the terminals. Similar to in DIFUSCO, we utilize the decoding scheme from CherryPick.

### J.2  ABLATION STUDIES

As the dimes only utilized a total of 360 instances, the initial concern is whether this quantity is enough to solve the problem. We conduct a more extended training involving 1920 iterations, with 5760 data instances, in 20 and 50 nodes. Table 12 shows the result of the extended training. The extended training leads to superior in-distribution results in both 20 and 50 nodes. The out-distribution results show contrasting patterns in the two settings, with superior performance shown in smaller nodes for STP20 and in bigger nodes for STP50.

## K  DECODING STRATEGIES IN THE STEINER TREE PROBLEM

### K.1  FEASIBLE SOLUTIONS IN THE STP

In neural network models for combinatorial optimization (CO) problems, a technique is needed to decode the neural network's raw output $G_\theta(s)$ into a feasible solution $\hat{f}_\theta(s) \in \mathcal{F}_s$ that meets the

Table 12: In- and Out-distribution Result on STP20 and STP50 for Extended Training

| Train Nodes | Train Instances | Test Nodes | | | | | |
|---|---|---|---|---|---|---|---|
| | | STP10 | STP20 | STP30 | STP50 | STP100 | STP200 |
| STP20 | 360 | $5.16 \pm 14.06$ | $6.86 \pm 14.10$ | $\mathbf{6.60 \pm 10.27}$ | $\mathbf{7.49 \pm 8.26}$ | $\mathbf{9.28 \pm 5.06}$ | $\mathbf{12.46 \pm 5.00}$ |
| | 5760 | $\mathbf{2.96 \pm 8.96}$ | $\mathbf{4.90 \pm 16.10}$ | $7.08 \pm 9.32$ | $11.70 \pm 9.51$ | $20.06 \pm 11.52$ | $40.66 \pm 25.44$ |
| STP50 | 360 | $\mathbf{6.51 \pm 17.09}$ | $\mathbf{8.27 \pm 16.81}$ | $\mathbf{9.59 \pm 13.63}$ | $11.17 \pm 10.11$ | $16.21 \pm 8.01$ | $23.80 \pm 11.01$ |
| | 5760 | $15.36 \pm 30.81$ | $16.52 \pm 33.21$ | $12.43 \pm 21.50$ | $\mathbf{5.45 \pm 8.09}$ | $\mathbf{10.75 \pm 8.42}$ | $\mathbf{9.46 \pm 4.05}$ |

specified constraints of CO problems. Unlike general large language models (LLMs), the constraints in CO problems are explicit and strict, defining feasible solutions for a given CO task. For example, in the Traveling Salesman Problem (TSP), a cycle that visits all nodes exactly once is considered feasible. In the Maximum Independent Set (MIS) problem, a solution must ensure no two selected nodes are adjacent. In the Steiner Tree Problem (STP), the solver must meet the following constraints:

1. All terminal nodes must be connected.

2. The selected edge set must not form any cycles to maintain a tree structure.

3. Redundant edges to connect the terminals must be excluded.

### K.2 GREEDY DECODING ALGORITHM IN THE STP

Various decoding strategies can satisfy the specified constraints, each influencing the performance of neural combinatorial optimization (NCO). To ensure a fair evaluation of the baselines in this study, we adapted a greedy decoding algorithm, similar to CherryPick (Yan et al., 2021), for the Steiner Tree Problem (STP) that adheres to the three constraints mentioned above. This algorithm is applied consistently across all baselines. The detailed pseudo-code for this algorithm is provided in Algorithm 2.

Our adapted greedy decoding approach for solving the STP is analogous to greedy decoding in LLMs, where edges are iteratively selected based on previously chosen edges. Initially, we select an arbitrary terminal and define a partial solution as a graph containing only this terminal. Subsequently, we define a candidate edge set as the edges adjacent to the current partial solution. Among these candidates, edges are selected based on scores derived from the raw output of the neural network, denoted as $G_\theta(s)$.

The decoding strategies slightly differ between autoregressive (AR) approaches, such as PtrNet (Meire et al., 2015), CherryPick (Yan et al., 2021), and AM (Kool et al., 2018), and non-autoregressive (nAR) approaches, such as DIFUSCO (Sun and Yang, 2023) and Dimes (Qiu et al., 2022). In nAR approaches, the values of $G_\theta(s)$ for the initial state $s$ are precomputed and are not be changed during the decoding process, and the edge scores are calculated from these values. Alternatively, in AR approaches, at each step of edge selection, the neural network receives the current partial solution as input, performs a forward pass to recalculate the node scores instead of directly computing the edge weights, and then selects the minimum weight edge that connects the partial solution graph to the remaining graph. To remove the redundant edges, we adapt the algorithm in (Kou et al., 1981).

---

**Algorithm 2** Decoding Algorithm for the STP

---

1: Initialize partial tree solution $G_p = \{V_p, E_p\}$ with the node set $V_p = \{v_t\}$ where $v_t \in T$ is a randomly selected terminal and the candidate edge set $E_p = \{(v, v') \,|\, (v, v') \in E, v \in V_p, v' \in V - V_p\}$
2: **while** $T \not\subset V_p$ **do**
3:     **if** non-autoregressive approaches **then**
4:         Choose the highest scored edge $(v_\star, v'_\star) := \arg\max_{e \in E} G_\theta(s, e)$ for $e \in E_p$
5:     **end if**
6:     **if** autoregressive approaches **then**
7:         Choose the highest scored node $v'_\star := \arg\max_{v' \in V - V_p} G_\theta(s, V_p, v)$
8:         Select the minimum weighted edge $(v_\star, v'_\star) = \arg\min_{e \in E_p} c_s(e)$
9:     **end if**
10:    Update the partial graph set $G_p$ with the selected node $v'_\star$ in $V_p$, i.e., $V_p = V_p \cup \{v'_\star\}$
11:    Update the selected edge $(v_\star, v'_\star)$ in the edge set $E_p$, i.e., $E_p = E_p \cup \{(v_\star, v'_\star)\}$
12: **end while**
13: Identify and remove redundant edges to enhance the efficiency of the network

---

