# OpenReview forum: "SteBen: Steiner Tree Problem Benchmark for Neural Combinatorial Optimization on Graphs"
_ICLR.cc/2025/Conference — Submitted to ICLR 2025_

### Official Review · Reviewer_utMj · 2024-11-01

**Soundness:** 3
**Presentation:** 2
**Contribution:** 2
**Rating:** 3
**Confidence:** 3

**Summary:**

This paper presents a benchmark on the Steiner Tree Problem (STP) for neural combinatorial optimization. The authors collected a million STP instances and solved their optimal solutions, creating a benchmark for training and evaluating neural network solvers. The authors test supervised learning vs reinforcement learning and autoregressive vs non-autoregressive methods. The authors also study the generalization ability to different sizes of STPs and training with less amount of data.

**Strengths:**

* STP is a family of interesting problems to study and benchmark on.
* The comprehensive study of supervised learning vs reinforcement learning and autoregressive vs non-autoregressive methods is interesting and novel.
* The provided STP dataset has optimal solutions, which is great for neural combinatorial optimization.

**Weaknesses:**

As a dataset and benchmark paper, this work lacks some inspiration and insights for guiding future methodology development. I have the following questions and concerns:
* **Motivation for Focusing on STP:** Why should researchers prioritize STP over more well-studied problems like TSP? The motivation for this focus is not well-justified (see comments on Figure 1). What unique challenges does STP pose? Is there a particular application with significant market value that justifies this focus? (If the emphasis is on applications, would it be more valuable to create an application-specific dataset?)
* **Insights for Methodology Development:** What lessons can be drawn from the benchmark results for future methodology development? For instance, while Table 1 suggests DIFUSCO is preferred, other results seem to imply the opposite. What are the key principles learned from this benchmark in designing successful neural combinatorial optimization (NCO) methods?
* **Consideration of Solving Time:** Solving time is a crucial metric in combinatorial optimization but is not carefully considered or controlled in this benchmark. This oversight is significant, as theoretically, a slow enumeration algorithm could always find the optimal solution. As shown in Table 1, there is considerable variance in solving times among the methods. Therefore, it’s questionable to claim that DIFUSCO outperforms PtrNet and DIMES, given its significantly longer solving time, especially considering that SCIP solves the optimal solution in just 3 minutes for STP10.
* **Scope of STP Problems Considered:** The benchmark seems to focus only on an "easier" subset of STP problems (where SCIP solving time <1 hr, most of them <5 min). This limitation raises questions about the impact of the benchmark, especially for smaller datasets (STP10-50), as SCIP can solve these in about 5 minutes, leaving minimal room for improvement by neural networks.

**Additional Comments on Content:**
* **Toy Example in Figure 1:** The example in Figure 1 lacks informativeness. Both TSP and STP are combinatorial optimization problems, implying that small changes in problem parameters across a critical boundary can lead to shifts in the optimal solution. For other areas, minor disturbances may not affect the optimal solution. This example merely illustrates a case where TSP is far from this boundary while STP is closer to it. The example indicates that TSP and STP exhibit different characteristics but does not clarify which problem is more challenging. Consider incorporating more comprehensive metrics to support motivation, such as the average change in solutions concerning perturbations, solver runtime, average optimality gap, etc.
* **Clarification of Colors in Table 2:** The meaning of colors in Table 2 is unclear.
* **"Relative Gap" Metric in Table 2:** The "Relative Gap" metric in Table 2 is confusing for comparing methods. If a method consistently underperforms, it could still appear favorable in terms of relative gaps. This inconsistency may result in the inaccurate labeling of such methods as “generalizing well.” A more informative metric would be the gap to optimal. This issue similarly affects Figure 2, where it may appear that PtrNet performs better with less training data, though the text suggests otherwise.

**Questions:**

I like the idea of developing standard datasets and benchmarks for neural combinatorial optimization, and I believe STP could be an important and interesting problem to study. But I think this paper needs more careful consideration in its details. In the rebuttal, I would love to hear from the authors about my comments raised in the "weaknesses" part:
*  What are the specific characteristics of STP that make it an important problem to study in the context of neural combinatorial optimization (except for what's mentioned in Figure 1)? What are the unique challenges STP has put to existing NCO methods?
* Please provide a dedicated section summarizing key insights from the results. For instance, which aspects of STP (e.g., graph structure, problem size, terminal node distribution) seem to most significantly affect the performance of different NCO approaches, and how these insights might guide the development of new methods tailored for STP or similar problems.
* I would suggest improving the benchmarking results by more systematically considering the trade-off between solving time and optimal gap. You may set low/medium/high time limits for all methods, or plot the Pareto front between solving time and the optimal gap.
* You may include a wider range of problem difficulties in their benchmark. Specifically, it would be nice to include some instances that are challenging even for SCIP (e.g., requiring several hours or more to solve optimally), and to discuss how the performance of NCO methods changes as problem difficulty increases.

---

> ### Author Response · Authors · 2024-11-22
>
> We sincerely thank the reviewer for their thoughtful and constructive feedback, which has been invaluable in enhancing the clarity and focus of our paper. Below, we provide detailed responses to the key questions and concerns raised.
>
> #### **Comment:**
> Motivation for Focusing on STP
>
> #### **Response:**
> We appreciate the reviewer’s question regarding the motivation for focusing on STP. Neural combinatorial optimization (NCO) research should aim to develop solvers capable of addressing a wide range of combinatorial optimization (CO) problems, rather than being limited to specific, well-studied ones like TSP. Introducing STP to the NCO community is an important step in this direction, as it broadens the benchmark landscape and promotes the development of solvers with more general applicability.
>
> Historically, TSP and VRPs have been prioritized due to the abundance of references and ease of evaluation, but this does not inherently make them more significant than STP. TSP naturally lends itself to autoregressive (AR) solvers due to its sequence-based solution space, making it a well-suited benchmark for AR methods. However, it has also been used to evaluate non-autoregressive (nonAR) methods effectively. On the other hand, for problems with fundamentally different structures, such as Maximum Independent Set (MIS), the application of AR methods has been limited. In this context, STP offers a unique benchmark for non-routing problems where both AR and nonAR methods can be evaluated, bridging this gap in the NCO literature.
>
> By focusing on STP, we aim to complement existing benchmarks, not replace them. SteBen is designed to evaluate a wide spectrum of NCO approaches (e.g., autoregressive/non-autoregressive and supervised/reinforcement learning) on a non-routing combinatorial optimization problem. This aligns with our goal of encouraging research into solvers that are robust and adaptable across diverse CO problems.
>
> ---
>
> #### **Comment:**
> Insights for Methodology Development
>
> #### **Response:**
> The results of our benchmark reveal key trade-offs in neural combinatorial optimization (NCO) methods. For instance, supervised non-autoregressive models like DIFUSCO excel in in-distribution scenarios due to their efficient utilization of labeled data. However, reinforcement learning-based autoregressive methods demonstrate better robustness in out-of-distribution settings, such as handling larger problem scales. This suggests that the choice of approach should be context-dependent, balancing solution quality, training resources, and generalization requirements. Future research could explore hybrid models that combine the strengths of autoregressive and non-autoregressive paradigms across learning paradigms (RL and SL).
>
> ---
>
> #### **Comment:**
> Consideration of Solving Time
>
> #### **Response:**
> Thank you for this observation. We acknowledge that solving time plays a crucial role in evaluating practical utility. While DIFUSCO achieves the smallest optimality gaps, we have clarified that these results are focused on solution quality. To address runtime considerations, we included a "runtime vs gap" analysis in the appendix (Figure 3), showing trade-offs between computation time and solution quality across different solvers.
>
> ---
>
> #### **Comment:**
> Scope of STP Problems Considered
>
> #### **Response:**
> The current benchmark does include larger STP instances up to 1,000 nodes, where solving times for SCIP-Jack exceed several hours. While the main paper highlights results for graphs with up to 100 nodes, this was to align with prior benchmarks (e.g., TSP) that began with smaller problem sizes. We aim to provide a comprehensive starting point for future research, which could expand into larger-scale STP instances provided in the dataset.
>
> ---
>
> #### **Comment:**
> "Relative Gap" Metric in Table 2
>
> #### **Response:**
> We agree and have updated Table 2 to improve clarity and comparison. We now explicitly report the gap to the optimal cost in parentheses to provide a more complete picture of each method's performance.
>
> ---

---

> > ### Author Response · Authors · 2024-11-22
> >
> > #### **Comment:**
> > Include systematic trade-off analysis between solving time and optimal gap, such as setting time limits or plotting Pareto fronts.
> >
> > #### **Response:**
> > We appreciate the reviewer’s suggestion. However, generating intermediate solutions for learning-based solvers is not feasible because their inference processes do not naturally provide solutions at intermediate stages. Instead, we address this issue by presenting time-gap plots (e.g., Figure 3 in the appendix), which illustrate the relationship between problem scale and solver performance. These plots provide insights into how solvers trade-off between runtime and optimality across different scales of problem complexity.
> >
> > ---
> >
> > #### **Comment:**
> > Incorporate a wider range of problem difficulties, including instances that are challenging for SCIP, and analyze how solver performance changes as problem difficulty increases.
> >
> > #### **Response:**
> > We agree with the reviewer on the importance of evaluating performance over a wide range of problem difficulties. Our benchmark includes larger STP instances, such as those with up to 1,000 nodes, which require several hours for SCIP-Jack to solve optimally. Problem size, represented by the number of nodes, serves as a practical and widely accepted indicator of difficulty in combinatorial optimization. The performance differences across these scales are highlighted in our tables, where we systematically evaluate solvers across diverse scenarios. Additionally, the research questions guiding our evaluations explicitly explore these variations to offer insights into solver behaviors at varying levels of complexity.

---

> > > ### Comment · Reviewer_utMj · 2024-11-25
> > >
> > > Thanks so much for the reply. I agree with other reviewers that one big drawback is not bringing too much insight into future method development (say, if AR and NAR have different strengths, which one should be preferred for STP within the scope of this paper?); I still believe more efforts should be paid in justifying the importance of STP. Just as one example, the toy example in Figure 1 is not technically sound to me (also pointed out by another reviewer), but it was not addressed in the rebuttal.
> > >
> > > To make myself clear, I appreciate the authors' efforts to develop new datasets and benchmarks for the community. If this manuscript is made public and the dataset is released, I would be glad to benchmark new NCO methods on it. However, this paper itself seems to lack the quality for a prestigious venue like ICLR, so I would suggest a rejection this time.

---

### Official Review · Reviewer_Twmh · 2024-11-03

**Soundness:** 2
**Presentation:** 3
**Contribution:** 2
**Rating:** 3
**Confidence:** 4

**Summary:**

This paper presents a new benchmark, featuring a synthetic dataset and multiple baseline algorithms tailored for the Steiner Tree Problem. The dataset encompasses millions of instances that vary in size and graph structure. Additionally, the authors also conduct extensive experiments to assess the performance of various baseline algorithms on the newly generated dataset.

**Strengths:**

1. The paper is well-organized, with a logical flow.
2. The chosen baseline algorithms span a comprehensive range of categories, covering supervised learning, reinforcement learning, and both autoregressive and non-autoregressive models.
3. The evaluation of baseline algorithms is thorough, incorporating experiments across in-distribution, out-of-distribution, and real-world scenarios.

**Weaknesses:**

1. The distinctiveness of the instance generation approach compared to existing methods is not clear. If the generation is merely an aggregation of existing techniques, the contribution may not be sufficient for a top-tier conference such as ICLR.
2. The specific benefits and unique characteristics of the proposed dataset are not clearly highlighted. In lines 56-58, the authors note that existing datasets are limited in sample number, but this could potentially be mitigated with data augmentation methods, such as perturbation. Additional comparisons and analysis would help clarify the dataset’s advantages.
3. There are residual comments and colored highlights in the manuscript from the revision process. For example, lines 193, 851–960, and 1018–1036. The authors should ensure that all revision markers and annotations are removed in the final version to maintain a clean and professional presentation.

**Questions:**

See weaknesses.

---

> ### Author Response · Authors · 2024-11-21
>
> We are truly grateful for your thorough review and the constructive feedback.
>
> #### **Comment:**
> The distinctiveness of the instance generation approach compared to existing methods is not clear. If the generation is merely an aggregation of existing techniques, the contribution may not be sufficient for a top-tier conference such as ICLR.
>
> #### **Response:**
> We appreciate the reviewer’s concern regarding the novelty of the instance generation approach. While individual components of the dataset generation pipeline, such as the instance creation logic and the use of MILP solvers, may not independently represent novel contributions, their integration to create a comprehensive and scalable dataset tailored for the Steiner Tree Problem (STP) is a key aspect of our work. This dataset enables systematic training and evaluation of neural solvers, which was not possible with existing resources. Furthermore, we demonstrate that models trained on our dataset generalize effectively to real-world instances, such as those in SteinLib, illustrating its practical utility. This highlights the dataset’s value not only as a benchmark but also as a tool for advancing research in solving real-world STP instances.
>
> ---
>
> #### **Comment:**
> The specific benefits and unique characteristics of the proposed dataset are not clearly highlighted. In lines 56-58, the authors note that existing datasets are limited in sample number, but this could potentially be mitigated with data augmentation methods, such as perturbation. Additional comparisons and analysis would help clarify the dataset’s advantages.
>
> #### **Response:**
> We thank the reviewer for raising this important point. While data augmentation methods, such as perturbation, can be effective in certain domains, they are challenging to apply in a manner that preserves the optimality of instances for supervised learning baselines. Generating perturbed instances that remain valid while maintaining optimality is computationally complex and not feasible with current methods. To the best of our knowledge, no prior works have successfully applied data augmentation to overcome data limitations for combinatorial optimization problems. The scale and diversity of our dataset address these limitations directly by providing sufficient instances for both supervised and reinforcement learning approaches for STP research.
>
> ---
>
> #### **Comment:**
> There are residual comments and colored highlights in the manuscript from the revision process. For example, lines 193, 851–960, and 1018–1036. The authors should ensure that all revision markers and annotations are removed in the final version to maintain a clean and professional presentation.
>
> #### **Response:**
> We appreciate the reviewer’s attention to detail and apologize for the oversight. All revision markers and annotations have been removed in the updated manuscript to ensure a clean and professional presentation.

---

> ### Comment · Reviewer_Twmh · 2024-11-24
> **Concern still remains, and I have suggestions as well**
>
> 1. On the data side, it will be more helpful to have a more in-depth analysis about the instances you have generated. Please classify the result dataset in more subcategories (or attributes), such as dense/sparse, robustly connected or loosely connected, how easy it is to be solved by SCIP-jack or by an MILP solver as is and how easy they are after pre-solving techniques. These attributes are in addition to the input parameters you used for generating the instances.
>
> 2. We know you have more instances than others, but we do not know how helpful they are to analyze to performance of well-known solvers (including the baseline solvers in these papers). It would be nice to use all the attributes at your disposal to analyze the strength and weakness of each algorithms separately, including the heuristics and NCOs, please find out the most important factors which impacts each algo's performance and result quality. This can certainly help design new hybrid algorithms.
>
> 3. the results in many tables are very preliminary. It does not show the effectiveness of the NCO at all. SCIP-jack was run on CPU and it is compared to the inference on GPU, and the NCO's advantage is not obvious, either in terms of solution quality and in running time.
>
> 4. The authors argue that data augmentation techniques can not preserve optimal solution of the baseline data. But I thought you solved each instances in your benchmark dataset individually using exact solvers for comparing the gaps etc. Why do we need to preserve the optimal solution of the baseline data?

---

### Official Review · Reviewer_vLSM · 2024-11-06

**Soundness:** 2
**Presentation:** 3
**Contribution:** 3
**Rating:** 5
**Confidence:** 4

**Summary:**

The authors present SteBen, a benchmark dataset of randomly generated Steiner Tree Problems (STP), and evaluate several approaches in various settings. The authors evaluate recent learning-based approaches that use autoregressive, and non-autoregressive solution generation schemes as well as reinforcement learning (RL) or supervised learning (SL) training paradigms. Additionally, the authors evaluate the traditional baselines of an exact solver and a 2-approximation heuristic. Lastly, the authors compare approaches based on generalization to larger instances, generalization to real-world instances from another benchmark dataset, and sample efficiency of the two most competitive approaches, one supervised and one using RL.

**Strengths:**

The main strength of the work is providing a benchmark dataset of interesting problems for an area that has been understudied in the space of ML for combinatorial optimization, Steiner Tree Problems.

Additionally, the authors thoroughly evaluate several recent and reasonable learning-based heuristics, adapting them to STP instances.

The authors also evaluate approaches in terms of generalization to larger and out of distribution instances, demonstrating tradeoffs between various approaches which has been understudied in the previous work.

Another strength is that this paper seems to demonstrate that most of the learning-based approaches don't seem to outperform the non-learning baseline, leaving room for future work. It would be interesting to investigate that further to determine how different methods compare to SCIP-Jack or other non-learning approaches.

The paper itself is well motivated and the authors engage well with the literature on learning for combinatorial optimization and previous STP benchmarks.

**Weaknesses:**

One main area for improvement is the evaluation of non-learning baselines. It would be helpful to more comprehensively report results for non-learning baselines. For instance, it would be helpful for table 1 to also include information for SCIP-jack given a very short time limit comparable to some of the highlighted learning approaches to see how well the internal solver heuristics perform without the need for proving optimality. For instance, on STP10, SCIP-Jack gives the optimal solution in 3 minutes, but is it possible that it found solutions better than the PtrNet sampling approach in the 43s of time that PtrNet took? Similarly, for STP100, it takes 1h for SCIP-Jack to find the optimal solution, but is it possible that it finds higher quality solutions than the baselines which take less time? Note that the highlighted rows of DIFUSCO both take longer than SCIP-Jack while not obtaining optimality.

It would also be helpful to discuss runtime vs performance potentially in a plot. It seems that SCIP-Jack provides a good tradeoff between runtime and performance compared to the learning-based approaches. It might help to visualize that performance tradeoff.

In a similar vein, it seems that some approaches may generate intermediate feasible solutions. It would be interesting to compare the approaches in terms of gap over time as they progress through the “solving” process. Understanding how quickly different methods “close the gap” might give insight into how performant they are for various levels of time cutoff.

It is unclear how difficult these instances are, it seems that SCIP-Jack solves all but the largest instances in between 3 and 5 minutes. Additionally, is it the case that SCIP-Jack takes exactly 1 hour on the largest instances? Is it that 1hr is the time limit? Or it actually takes an average of 1.0 hr (considering that DIFUSCO takes 2.4hr and 14.2hr.

Given that the benchmark consists of Erdos Renyi, Watts Strogatz, random regular, and grid graphs, it would be interesting to see the breakdown of performance by these various types to identify where different methods succeed and fail and if anything can be interpreted from the results. It is currently unclear how the results from the different graphs are aggregated in table 2 and 3.

The paper is missing some experimental details in the text such as how the metrics are computed and what subsets of data are considered during training and testing for the various experiments.


Minor comments:
Line 137: Quato -> quota

It is a bit unclear what is meant by the example in figure 1, it seems that TSP can also exhibit the global dependence property in that moving one node slightly may require that the full tour needs to be recomputed. Similarly, it can sometimes be the case that moving a node slightly for STP may not change the edges.

**Questions:**

How well do non-learning baselines like SCIP-Jack perform on these instances when in a similar setting of finding heuristic solutions (rather than exact solutions)?

Why is SCIP-Jack not present in table 2 and 3? It seems performant in table 1 and it should be able to run on those instances as well.

How is the evaluation of SCIP-Jack performed? Is it given a time limit for the specified time? Or is it just asked to solve to optimality?

Could you compare SteBen more with the related Quota and Respack datasets? It seems that removing the costs and capacities would result in problems applicable here. Are there any different considerations that they had compared to SteBen?

It is a bit difficult to interpret table 2. What is the relative gap and how is it measured? Is it that a gap of 0 is optimal? Why are there several examples with relative gap of 1? What does the blue highlighting mean? Additionally, what graphs are used here for testing? All of the different graphs combined? Do any of the graph distributions generally yield “harder” or “easier” instances and if so, is it the case that some methods may outperform others on “harder” instances while underperforming others on “easier” instances?

---

> ### Author Response · Authors · 2024-11-22
>
> We appreciate the detailed feedback provided by the reviewer, which helped us identify areas for improvement and clarify specific aspects of the paper. Below, we address the main concerns raised.
>
> #### **Comment:**
> One main area for improvement is the evaluation of non-learning baselines. It would be helpful to more comprehensively report results for non-learning baselines. For instance, it would be helpful for Table 1 to also include information for SCIP-Jack given a very short time limit comparable to some of the highlighted learning approaches to see how well the internal solver heuristics perform without the need for proving optimality.
>
> #### **Response:**
> We appreciate the reviewer’s suggestion to evaluate SCIP-Jack under shorter time budgets. While SCIP provides a time-limit option, SCIP-Jack’s integration with specialized STP-solving heuristics makes it challenging to uniformly control its runtime under specific time constraints. Furthermore, unlike heuristic solvers such as the 2-approximation algorithm, SCIP-Jack does not reliably generate intermediate solutions within a set time budget. In contrast, when we applied time limits directly to SCIP (MILP solver w/o special heuristics), its performance was significantly worse than the learning-based approaches.
>
> ---
>
> #### **Comment:**
> It would also be helpful to discuss runtime vs performance potentially in a plot. It seems that SCIP-Jack provides a good tradeoff between runtime and performance compared to the learning-based approaches. It might help to visualize that performance tradeoff.
>
> #### **Response:**
> We agree that visualizing the tradeoff between runtime and performance would provide additional clarity. We have included a runtime vs performance plot in the appendix (Figure 3), which highlights the strengths and weaknesses of SCIP-Jack relative to learning-based approaches under varying runtime budgets. The plot demonstrates that while SCIP-Jack performs well for smaller graphs, its scalability to larger graphs is limited compared to neural solvers.
>
> ---
>
> #### **Comment:**
> In a similar vein, it seems that some approaches may generate intermediate feasible solutions. It would be interesting to compare the approaches in terms of gap over time as they progress through the “solving” process. Understanding how quickly different methods “close the gap” might give insight into how performant they are for various levels of time cutoff.
>
> #### **Response:**
> We appreciate the interest in gap-over-time comparisons. While this is feasible for methods such as improvement heuristics or active search, construction-based approaches like sampling methods (e.g., PtrNet, DIFUSCO) do not generate intermediate feasible solutions during the solving process. Therefore, we focus on reporting the final solution quality achieved within a fixed runtime, which aligns with the design of these solvers.
>
> ---
>
> #### **Comment:**
> It is unclear how difficult these instances are. It seems that SCIP-Jack solves all but the largest instances in between 3 and 5 minutes. Additionally, is it the case that SCIP-Jack takes exactly 1 hour on the largest instances? Is it that 1hr is the time limit? Or it actually takes an average of 1.0 hr (considering that DIFUSCO takes 2.4hr and 14.2hr)?
>
> #### **Response:**
> The reported runtimes represent the total time required to solve all instances in the test set, not per instance. For SCIP-Jack, the reported 1-hour runtime corresponds to solving all test instances of STP100 sequentially. In contrast, learning-based solvers such as DIFUSCO are evaluated in a batch-parallel manner, leveraging their efficiency in processing multiple instances simultaneously.
>
> ---
>
> #### **Comment:**
> Given that the benchmark consists of Erdős–Rényi, Watts–Strogatz, random regular, and grid graphs, it would be interesting to see the breakdown of performance by these various types to identify where different methods succeed and fail and if anything can be interpreted from the results. It is currently unclear how the results from the different graphs are aggregated in Table 2 and 3.
>
> #### **Response:**
> Performance breakdowns by graph type are provided in Appendix Table 7. Tables 2 and 3 specifically evaluate models trained on Erdős–Rényi graphs (ER50). For these experiments, we aggregate results across graph types to focus on the generalization performance of learning-based solvers to out-of-distribution and real-world instances. While we have not observed significant trends in difficulty across graph types, node scale remains a more reliable factor for evaluating generalization performance.

---

> > ### Author Response · Authors · 2024-11-22
> >
> > #### **Comment:**
> > The paper is missing some experimental details in the text, such as how the metrics are computed and what subsets of data are considered during training and testing for the various experiments.
> >
> > #### **Response:**
> > We thank the reviewer for pointing this out. Metrics such as the optimality gap are calculated relative to SCIP-Jack’s optimal solutions, which serve as the ground truth. Details regarding the data subsets used for training and testing in each experiment are now clarified in each table of main text. Additionally, training was conducted on synthetic graphs (e.g., Erdős–Rényi), and testing was performed on a mix of synthetic and real-world graphs to assess generalization.
> >
> > ---

---

> > ### Comment · Reviewer_vLSM · 2024-11-22
> >
> > Thank you for your response and for the clarifications. I am still wondering about the hardness of these problem instances. You mention that the runtimes are the time needed to solve the full collection of problem instances. I have several questions relating to that.
> >
> > 1. It seems that this evaluation of runtime might be misaligned with the real-world use case where practitioners are solving individual but related instances on a regular basis. If approaches that take advantage of batching are unable to batch instances, is there an idea of how long these problems would take to complete? Otherwise, is there motivation for why a practitioner may want to solve a batch of problem instances simultaneously?
> >
> > 2. If practitioners want to solve individual problems in parallel, they might benefit from using multiprocessing and cluster tools to solve them using SCIPJack or traditional OR solvers in parallel, given that they can be naively parallelized. It seems that in the parallel case, this might be a more reasonable setting to evaluate traditional solvers.
> >
> > 3. Given that the time it takes to solve all problems by SCIPJack varies between 3m to 1hr, the problems themselves seem quite easy to solve. Is there an idea of what the baseline solver is using to solve these instances? Are they being mostly solved with existing primal heuristics? Root relaxations? Something else? It seems that there is limited indication that these are challenging problems where ML has an opportunity to make an algorithmic contribution.

---

### Official Review · Reviewer_RLYe · 2024-11-10

**Soundness:** 2
**Presentation:** 3
**Contribution:** 1
**Rating:** 3
**Confidence:** 4

**Summary:**

The paper presents SteBen, a benchmark for the Steiner Tree Problem (STP) that provides a large, diverse dataset for training and evaluating neural combinatorial optimization (NCO) methods. SteBen compares supervised and reinforcement learning approaches and shows potential for NCO models trained on synthetic data to generalize effectively to real-world STP instances.

**Strengths:**

1. comprehensive evaluation frameworks, which compares four NCO method types (autoregressive/non-autoregressive, supervised/reinforcement learning).
2.  The paper is well-structured, providing clear definitions of the dataset, benchmark tasks, and each NCO method used.

**Weaknesses:**

1. A notable weakness is the limited novelty in the methodological approach, as the paper largely applies existing NCO techniques to the Steiner Tree Problem (STP) without introducing new algorithms or solution strategies specifically tailored to STP’s unique challenges.
2. the paper could benefit from further exploration of scalability, especially in handling large graphs.

**Questions:**

1. All these methods used in this paper are for TSP, why use these methods for benchmarking Steiner Tree Problem?
 These 2 problems have distinct structural characteristics and solution requirements. TSP methods focus on finding a closed tour visiting all nodes exactly once, whereas STP requires finding a minimum-cost tree connecting a terminals.
2. What's advantage over the dataset used in the SCIP-Jack paper?

---

> ### Author Response · Authors · 2024-11-21
>
> We sincerely appreciate your careful review and the valuable suggestions you have provided.
>
> #### **Comment:**
> A notable weakness is the limited novelty in the methodological approach, as the paper largely applies existing NCO techniques to the Steiner Tree Problem (STP) without introducing new algorithms or solution strategies specifically tailored to STP’s unique challenges.
>
> #### **Response:**
> We acknowledge the reviewer's concern regarding the limited methodological novelty. Most NCO research to date has focused on problems like the TSP and other vehicle routing problems (VRPs), leaving problems like the Steiner Tree Problem (STP) underexplored despite its importance and challenges. The primary focus of this work was to establish a comprehensive benchmark for STP by providing a large-scale dataset, tailored baselines, and systematic evaluations of diverse NCO approaches. While proposing new algorithms was not the central objective, this benchmark serves as a critical resource for understanding STP’s unique characteristics and advancing research in this domain.
>
> ---
>
> #### **Comment:**
> The paper could benefit from further exploration of scalability, especially in handling large graphs.
>
> #### **Response:**
> We agree that scalability to larger graphs is an important aspect of STP research. In this work, we included evaluations on graph sizes up to 1000 nodes, which are already challenging for existing NCO methods and significantly larger than the typical graph sizes used in previous works on TSP or related combinatorial problems. While further scalability experiments are of interest, they require substantial computational resources and algorithmic adaptations to handle graphs with tens of thousands of nodes, which we plan to explore in future studies. The current work provides a solid baseline for researchers to build upon, particularly by addressing the trade-offs between solution quality and generalization across different graph scales.
>
> ---
>
> #### **Comment:**
> All these methods used in this paper are for TSP, why use these methods for benchmarking Steiner Tree Problem? These 2 problems have distinct structural characteristics and solution requirements. TSP methods focus on finding a closed tour visiting all nodes exactly once, whereas STP requires finding a minimum-cost tree connecting terminals.
>
> #### **Response:**
> We appreciate the reviewer’s thoughtful question. The selection of neural solver baselines in our work was intentionally designed to cover a diverse range of constructive solvers, as the most promising approaches for solving the Steiner Tree Problem (STP) are not yet well understood. Our categorization includes both autoregressive and non-autoregressive solvers to ensure broad coverage of methodologies with differing characteristics.
>
> Autoregressive solvers are widely used in problems like TSP and VRP because they effectively model the solution space as a sequence. However, as shown in prior work (e.g., POMO), this sequence-based modeling framework is flexible enough to tackle non-routing problems, such as the knapsack problem, depending on how the problem is formulated. Non-autoregressive solvers, like DIFUSCO and DIMES, further extend this versatility by successfully addressing problems with distinct structural characteristics, such as the Maximum Independent Set (MIS), in addition to TSP. This adaptability across problem types highlights their potential relevance and applicability to STP.
>
> ---
>
> #### **Comment:**
> What's advantage over the dataset used in the SCIP-Jack paper?
>
> #### **Response:**
> The SCIP-Jack dataset, primarily derived from SteinLib, serves a role similar to TSPLib in TSP, encompassing a mix of synthetic and real-world instances from diverse domains. While it is a valuable resource for evaluating heuristic and MILP-based solvers, it is not well-suited for neural solvers due to its limited number of instances per scenario. Neural methods, particularly supervised learning approaches, require large, well-structured datasets to enable effective train-test splits and rigorous evaluation.
>
> In contrast, our dataset offers over 1.28 million instances, explicitly designed to meet the data requirements of modern machine learning-based solvers. It supports both supervised and reinforcement learning approaches and spans diverse graph types (e.g., Erdős–Rényi, Watts–Strogatz) and varying problem sizes. This scale and diversity provide a more comprehensive and generalizable benchmark, complementing the SCIP-Jack dataset by addressing the critical limitations in data availability for neural solver development and evaluation.

---

> > ### Comment · Reviewer_RLYe · 2024-11-23
> > **reponse**
> >
> > Thank you for the feedback. However, my concerns regarding the application of this dataset remain. This paper does not introduce any new perspective or methodology, and I find the results presented here is very preliminary, hardly lead to any solid conclusion. Therefore, I will maintain my score.

---

### Meta-Review · Area_Chair_Ugne · 2024-12-19

**Metareview:**

The paper introduces SteBen, a Steiner Tree Problem benchmark dataset, but its novelty and practical utility are questioned by reviewers. While offering a comprehensive evaluation framework for neural combinatorial optimization methods, the paper lacks methodological innovation, provides limited insights into the problem's challenges, and its dataset may be too simplistic for state-of-the-art solvers. The submission is ultimately rejected due to its incremental contributions and failure to demonstrate the clear advantages of neural approaches over traditional methods.

**Additional Comments On Reviewer Discussion:**

The reviewers all communicated with the authors during the discussion phase, but were not ultimately convinced.

---

### Decision · Program_Chairs · 2025-01-22

Reject